# Tokenisation is NP-Complete

Philip Whittington [1]   Gregor Bachmann [1]   Tiago Pimentel [1]

## Abstract

In this work, we prove the NP-completeness of two variants of tokenisation, defined here as the problem of compressing a dataset to at most $\delta$ symbols by either finding a vocabulary directly (*direct* tokenisation), or selecting a sequence of merge operations (*bottom-up* tokenisation).

## 1. Introduction

Tokenisation is at the heart of natural language processing (NLP) being the first step required to use a language model. Given a string of **characters c**, a tokeniser converts it into a string of **subwords s**. Language models are then trained to estimate distributions over subword-strings—never seeing the original character-strings. Despite its prominent role, however, much remains unknown about tokenisation. We still do not know, for instance, what makes a good tokeniser (Gowda & May, 2020; Cognetta et al., 2024): which characteristics should its produced subwords **s** have to be a good starting point for language modelling? If we knew this, then we could define an **objective function** with which we could evaluate tokenisers.

Another open question is how to—given such an objective function—efficiently find a tokeniser which maximises it. Byte pair encoding (BPE; Gage, 1994; Sennrich et al., 2016), for instance, is a greedy solution to find a tokeniser which maximises a text's compression. UnigramLM (Kudo, 2018) is a heuristic method to find a tokeniser that maximises its tokenised text's unigram log-probability. Both these methods, however, are approximate: they do not necessarily find an optimal tokeniser according to their objective function. This raises the question of whether finding such optimal tokenisers efficiently is even possible.

In this paper, we answer this question (at least partially), proving the NP-completeness of several variants of this tokenisation problem. Specifically, we focus on finding

[1]Department of Computer Science, ETH Zurich, Zurich, Switzerland. Correspondence to: Philip Whittington <philip.whittington@inf.ethz.ch>, Tiago Pimentel <tiago.pimentel@inf.ethz.ch>.

*Non-archival presentation at ICML 2025 Tokenization Workshop (TokShop)*, Vancouver, Canada. 2025.

tokenisers that maximise the **compression** of a text.[1] Given this objective, we then define the **tokenisation problem** as the task of finding a tokeniser which compresses a dataset to at most $\delta$ symbols. Notably, prior work imposes different constraints on how tokenisers are defined; here we consider two variants. In **direct tokenisation**, the desired compression must be reached by choosing a vocabulary (i.e., a set of subwords) which is directly used to represent the text. In **bottom-up tokenisation**, the desired compression must be reached by finding a sequence of merge operations instead, which we apply to the input text.

We prove the NP-hardness of both of these tokenisation problems (as well as of some variants thereof) by reducing from the **max 2-satisfiability** problem.[2] Practically speaking, our results imply that we are unlikely to discover an efficient algorithm for the problem of finding optimal tokenisers, and that we should focus on approximate algorithms (such as BPE or UnigramLM) instead.

## 2. How to Choose a Tokeniser?

Given a tokeniser, any character-level distribution has an equivalent subword-level distribution (Pimentel & Meister, 2024; Phan et al., 2024; Giulianelli et al., 2024). Ergo, despite the distribution we may wish to language model, a sufficiently expressive model should be able to represent it exactly; this is true regardless which tokeniser is used. In theory, thus, a researcher's choice of tokeniser should not influence their language model's quality.

In practice, however, a bad choice of tokeniser can have undesirable effects on downstream applications. For instance, performing standard arithmetic tasks (e.g., $317 + 421$) can be difficult even for large models (Nogueira et al., 2021; Muffo et al., 2022) due to the arbitrary splitting of numbers into subwords. Indeed, simple changes in how numbers are

---

[1]The compression achieved by a tokeniser correlates with downstream language modelling performance (Gallé, 2019; Zouhar et al., 2023a) and computational efficiency.

[2]We note two related concurrent works. Kozma & Voderholzer (2024) also prove the NP-completeness of bottom-up tokenisation; in fact, they prove something stronger: its APX-hardness. Lim et al. (2025) prove the NP-completeness of a restricted variant of direct tokenisation, in which a set of candidate tokens is previously specified.

tokenised can improve performance in such tasks (Singh & Strouse, 2024). Similar issues arise when prompting language models to count letters, where even advanced models such as GPT-4 infamously cannot correctly count the number of occurrences of the letter r in the word strawberry.

This raises the question of how to select a good tokeniser. Ideally, we would choose the tokeniser which maximises downstream language modelling performance. Unfortunately, we do not know how to measure such performance without fully training a model, making its direct maximisation computationally infeasible. Rather, we thus optimise proxy objectives—assumed to correlate with downstream performance. Among these are unigram log-probability (Kudo, 2018), Rényi efficiency (Zouhar et al., 2023a), and compression (Gallé, 2019).

We focus on compression in this paper. Denoting our tokenisation's **objective function** as $\mathfrak{G}$, we write this objective as: $\mathfrak{G}(\mathbf{s}) = -|\mathbf{s}|$. Improved compression leads to: (i) more efficient training and inference, due to shortened inputs;[3] (ii) improved downstream performance, at least to a certain extent (Gallé, 2019; Rust et al., 2021; Zouhar et al., 2023a; Goldman et al., 2024);[4] and (iii) fairer multilingual treatment—assuming similar compression across languages—given models' limited context lengths and the per-token costs to use proprietary models (Petrov et al., 2023; Ahia et al., 2023).

---

> **Our Notation's Colour-coding**
>
> - Blue for raw data (i.e., characters $\mathbf{c} \in \Sigma^*$);
> - Magenta for tokeniser-specific data (i.e., subwords $\mathbf{s} \in \mathcal{S}^*$ and merges $\mathbf{m} \in \mathcal{M}^*$);
> - Orange for functions (e.g., tok).

---

## 3. Defining a Tokeniser

A tokeniser can be defined as a 3-tuple $\langle \mathcal{S}, \texttt{tok}, \texttt{detok} \rangle$, composed of a vocabulary, a tokenisation and a detokenisation function. Before defining these terms, however, we require some notation. Let $\mathbf{c} \in \Sigma^*$ be a **character-string**,[5] i.e., a sequence of characters $c$ from alphabet $\Sigma$, which we write as $\mathbf{c} = c_1 c_2 \cdots c_{|\mathbf{c}|}$. Further, let $\mathcal{D} = \{\mathbf{c}_n\}_{n=1}^{N}$ be a dataset

of character-strings.[6] A subword $s \in \mathcal{S}$ represents a non-empty character-string $\mathbf{c}$ (where sequence $\mathbf{c}$ can have length one). Finally, let $\mathbf{s} \in \mathcal{S}^*$ be a **subword-string**. Just like a single subword, a subword-string $\mathbf{s} = \langle s_1, s_2, \cdots, s_{|\mathbf{s}|} \rangle$ represents a character-string via the concatenation of its subwords' characters:

$$\texttt{concat}(\mathbf{s}) = s_1 \circ s_2 \circ ... \circ s_{|\mathbf{s}|} \quad (1)$$

and we say that a pair of character and subword strings are equivalent if:

$$\mathbf{c} \overset{\circ}{=} \mathbf{s} \iff \mathbf{c} = \texttt{concat}(\mathbf{s}) \quad (2)$$

Given the notation above, we can now define the items in tuple $\langle \mathcal{S}, \texttt{tok}, \texttt{detok} \rangle$. A tokeniser's **vocabulary** is a set of subwords $\mathcal{S} \subset \Sigma^+$ such that $\Sigma \subseteq \mathcal{S}$;[7] we say its size is $|\mathcal{S}| = |\Sigma| + K$. Further, a **detokenisation function** is defined as $\texttt{detok} : \mathcal{S}^* \to \Sigma^*$ and given a subword-string it outputs the character-string it represents. This function thus is simply defined as $\texttt{detok}(\mathbf{s}) \overset{\text{def}}{=} \texttt{concat}(\mathbf{s})$.

Finally, we are left with defining a **tokenisation function** $\texttt{tok} : \Sigma^* \to \mathcal{S}^*$, which maps from character- to subword-strings. Notably, these functions always ensure the equivalence $\mathbf{c} \overset{\circ}{=} \mathbf{s}$ for $\mathbf{s} = \texttt{tok}(\mathbf{c})$. Several tokenisation functions, however, are compatible with this constraint, as given a vocabulary, many subword-strings may be equivalent to the same character-string. For instance, given $\mathcal{S} = \{a, c, t, at\}$, the string $\mathbf{c} = c, a, t$ could be tokenised as $\mathbf{s} = \langle c, a, t \rangle$ or as $\mathbf{s} = \langle c, at \rangle$. Most researchers define tokenisation functions in one of two ways, which we term direct and bottom-up tokenisation functions; we define these next.

### 3.1. Direct Tokenisation Functions

In direct tokenisation, a character-string is directly replaced by an optimal subword-string. To implement this, one must thus first define what *optimal* means; this is done through an objective function $\mathfrak{G}$ which, given a subword-string, returns a score. Given a previously chosen vocabulary $\mathcal{S}$ (we discuss how to find $\mathcal{S}$ in §5), a direct tokenisation function then encodes string $\mathbf{c}$ as:

$$\texttt{tok}_{\diamond}[\mathcal{S}](\mathbf{c}) = \underset{\mathbf{s} \in \mathcal{S}^*}{\arg\max} \, \mathfrak{G}(\mathbf{s}) \quad (3)$$

$$\text{s.t. } \mathbf{s} \overset{\circ}{=} \mathbf{c}$$

---

[3]Recent work tries to improve the computational efficiency of byte-level models (Yu et al., 2023; Pagnoni et al., 2024).

[4]Although, see Ali et al. (2024), who argue that compression might be a necessary but not sufficient condition for good downstream performance, and Schmidt et al. (2024), who argue that compression and downstream performance have a more complex relationship than prior work suggests.

[5]We note that $\Sigma^*$ denotes the Kleene star of $\Sigma$ (i.e., $\cup_{i=0}^{\infty} \Sigma^i$), and $\Sigma^+$ denotes its Kleene plus (i.e., $\cup_{i=1}^{\infty} \Sigma^i$).

[6]We note that we use set notation here, but our datasets are actually multisets—datasets can include the same string $\mathbf{c}$ multiple times. We show that tokenisation is still NP-complete for datasets with no repetitions in §6.3. Further, we impose no constraint on the kind of string present in these datasets: each $\mathbf{c}_n$ can be either a raw or pre-tokenised character-string (i.e., either a full document or a whitespace-separated word).

[7]$\Sigma \subseteq \mathcal{S}$ is typically enforced to guarantee that every $\mathbf{c} \in \Sigma^*$ can be represented by at least one subword-string $\mathbf{s} \in \mathcal{S}^*$.

In words, given a vocabulary $\mathcal{S}$, function $\texttt{tok}_\Diamond$ returns the optimal subword-string $\mathbf{s} \in \mathcal{S}^*$ which is equivalent to the input character-string $\mathbf{c}$. We then set $\texttt{tok}(\mathbf{c}) \overset{\text{def}}{=} \texttt{tok}_\Diamond[\mathcal{S}](\mathbf{c})$. Different choices of $\mathfrak{G}$ recover methods such as UnigramLM's tokenisation function (Kudo, 2018) or PathPiece (Schmidt et al., 2024). Notably, in general, this function is not efficiently computable.[8]

In this paper, we are concerned with tokenisers that use compression as their objective: that is, for which $\mathfrak{G}(\mathbf{s}) = -|\mathbf{s}|$. In this case, we can rewrite the direct tokenisation function as:

$$\texttt{tok}_\Diamond[\mathcal{S}](\mathbf{c}) = \underset{\mathbf{s} \in \mathcal{S}^*}{\arg\min} \, |\mathbf{s}| \tag{4}$$
$$\text{s.t. } \mathbf{s} \overset{\circ}{=} \mathbf{c}$$

Importantly, in the case of compression, this equation can be computed efficiently (as shown in §5.1).

### 3.2. Bottom-up Tokenisation Functions

In bottom-up tokenisation, one starts with a set of character-strings, and merges their symbols bottom-up, one pair at a time.[9] Formally, let $m \in \mathcal{M}$ be a **merge**, defined as a pair of subwords: $m = \langle s_1, s_2 \rangle$. Further, let $\mathcal{M} \overset{\text{def}}{=} \Sigma^+ \times \Sigma^+$. Now, let $\texttt{merge}$ be a functional; given merge $m = \langle s_1, s_2 \rangle$, it returns a function $\texttt{merge}[m] : \mathcal{S}^* \to (\mathcal{S} \cup \{s_1 \circ s_2\})^*$ which operates on string $\mathbf{s}$ left-to-right, replacing every occurrence of $s_1$ followed by $s_2$ in it with subword $s' = s_1 \circ s_2$. E.g., given $\mathbf{s} = \langle wo, r, ld \rangle$ and $m = \langle wo, r \rangle$, the output of $\texttt{merge}[m](\mathbf{s})$ is $\langle wor, ld \rangle$.

Consider now $\mathbf{m} \in \mathcal{M}^*$, a sequence of merges. Given a character-string $\mathbf{c} \in \Sigma^*$, a bottom-up tokenisation function compresses it as:

$$\texttt{tok}_\uparrow[\mathbf{m}](\mathbf{c}) = \left( \bigodot_{z=1}^{|\mathbf{m}|} \texttt{merge}[m_z] \right)(\mathbf{c}) \tag{5}$$

where $\bigodot$ represents function composition, e.g., $\bigodot_{z=1}^2 \texttt{merge}[m_z] = \texttt{merge}[m_2] \odot \texttt{merge}[m_1]$. Bottom-up tokenisers then set $\texttt{tok} \overset{\text{def}}{=} \texttt{tok}_\uparrow[\mathbf{m}]$. Further, a merge sequence $\mathbf{m}$ is also used to set a bottom-up tokeniser's vocabulary as:

$$\mathcal{S} = \Sigma \cup \{s_1 \circ s_2 \mid \langle s_1, s_2 \rangle \in \mathbf{m}\} \tag{6}$$

where $|\mathbf{m}| = K$ implies this vocabulary has size $|\mathcal{S}| = |\Sigma| + K$, as before.

---

[8]In fact, Geh et al. (2024) shows that it is NP-complete for $\mathfrak{G}(\mathbf{s}) = \sum_{t=1}^{|\mathbf{s}|} \log p_{\boldsymbol{\theta}}(s_t \mid \mathbf{s}_{<t})$, where $p_{\boldsymbol{\theta}}$ is a language model.

[9]Currently, this is likely the most common tokenisation function, being used in popular tokenisers such as, e.g., GPT-4's (OpenAI, 2023), LLaMA's (Touvron et al., 2023a;b), and Pythia's (Biderman et al., 2023).

## 4. Maximum 2-Satisfiability

Our paper's goal is to prove the NP-completeness of tokenisation. To show this, we must reduce an NP-hard problem to tokenisation in polynomial time. We will rely on the **maximum 2-satisfiability** problem ($\texttt{max-2-SAT}$) for this, whose definition we provide here. The NP-hardness of $\texttt{max-2-SAT}$ was proven by Garey et al. (1974).

**Definition 1.** *Let $\mathcal{X} = \{X_j\}_{j=1}^J$ be a set of variables; each of these variables are assigned values $x_j \in \{\texttt{F}, \texttt{T}\}$, and we write $\chi = \{x_j\}_{j=1}^J \in \{\texttt{F}, \texttt{T}\}^J$. Let $\mathcal{C} = \{(L_i^1 \vee L_i^2)\}_{i=1}^I$ be a set of clauses,[10] where each literal $L$ represents either a variable $X_j$ or its negation $\neg X_j$. The $\texttt{max-2-SAT}$ decision problem requires deciding whether there exists an assignment for which at least $\gamma$ clauses are satisfied:*

$$\gamma \leq \max_{\chi \in \{\texttt{F}, \texttt{T}\}^J} \sum_{i=1}^I \mathbb{1}_\chi \{L_i^1 \vee L_i^2\} \tag{7}$$

*where $\mathbb{1}_\chi$ is an indicator function which evaluates the clause and returns one if the clause is satisfied by $\chi$ and zero otherwise.*

For mathematical convenience, we will write $\text{M2S}(\mathcal{X}, \mathcal{C}, \gamma)$ for a function which returns $\texttt{T}$ if its input is satisfiable under a $\texttt{max-2-SAT}$ decision problem, and $\texttt{F}$ otherwise. As a concrete example, consider the set of variables $\mathcal{X} = \{X_1, X_2\}$ and the set of clauses $\mathcal{C} = \{X_1 \vee X_2, \neg X_1 \vee X_2, X_1 \vee \neg X_2, \neg X_1 \vee \neg X_2\}$. The assignment $x_1 = \texttt{T}, x_2 = \texttt{T}$ leads to 3 clauses being satisfied, which is the optimum. For this example, we thus have that $\text{M2S}(\mathcal{X}, \mathcal{C}, 3) = \texttt{T}$, but that $\text{M2S}(\mathcal{X}, \mathcal{C}, 4) = \texttt{F}$.

## 5. Finding Optimal Direct Tokenisers

We are now left with the task of finding an optimal tokeniser. We do this by selecting either: its vocabulary in direct tokenisation, since $\texttt{tok} = \texttt{tok}_\Diamond[\mathcal{S}]$; or its merge sequence in bottom-up tokenisation, since $\texttt{tok} = \texttt{tok}_\uparrow[\mathbf{m}]$ and since its vocabulary is chosen according to Eq. (6). (Note that in §3, we only showed how to apply tokenisers at inference time, but not how to find them.) In this section, we focus on direct tokenisation, defining its optimisation and decision problems; we then prove its NP-completeness. The optimisation problem is defined as follows.

**Definition 2.** *Given a dataset $\mathcal{D}$ and a vocabulary size $K$, the **direct tokenisation optimisation problem** is to find a*

---

[10]$\texttt{max-2-SAT}$ also allows clauses to have a single literal $L_i$. In this case, we can always rewrite the clause as $(L_i \vee L_i)$ with no change to the solution of this decision problem.

*vocabulary* $\mathcal{S}_{\mathrm{opt}} \subset \Sigma^+$ *which maximally compresses* $\mathcal{D}$:

$$\mathcal{S}_{\mathrm{opt}} = \arg\min_{\mathcal{S} \subset \Sigma^+} \sum_{\mathbf{c} \in \mathcal{D}} |\mathtt{tok}_\diamond[\mathcal{S}](\mathbf{c})| \qquad (8)$$
$$\text{s.t. } |\mathcal{S}| = |\Sigma| + K$$

We can similarly define direct tokenisation's decision problem.

**Definition 3.** *Given a dataset* $\mathcal{D}$ *and a vocabulary size* $K$*, the **direct tokenisation decision problem** requires deciding whether there exists a vocabulary* $\mathcal{S} \subset \Sigma^+$ *which compresses* $\mathcal{D}$ *to at most* $\delta$ *symbols:*

$$\delta \geq \min_{\mathcal{S} \subset \Sigma^+} \sum_{\mathbf{c} \in \mathcal{D}} |\mathtt{tok}_\diamond[\mathcal{S}](\mathbf{c})| \qquad (9)$$
$$\text{s.t. } |\mathcal{S}| = |\Sigma| + K$$

We write $\mathrm{Tok}_\diamond(\mathcal{D}, K, \delta)$ for a function which returns T if a direct tokenisation decision problem with those inputs is satisfiable, and F otherwise. Note that, whenever $|\mathcal{D}| \leq K$, the solution to the problem above is trivial, as an optimal solution simply requires including all strings $\mathbf{c}_n$ in vocabulary $\mathcal{S}$. As we show next, however, in the general case the above decision problem is NP-complete. We now state this as a theorem, which we will prove in the next two sections.

**Theorem 1.** *The direct tokenisation decision problem, as in Definition 3, is NP-complete.*

*Proof.* A decision problem is considered to be NP-complete if: (i) it is in NP; (ii) it is NP-hard. We prove these conditions in §5.1 and §5.2. $\square$

## 5.1. Direct Tokenisation is in NP

A decision problem is in the nondeterministic polynomial time class (NP) if, given a **certificate** of polynomial length, one can verify that certificate in polynomial time. Specifically, a certificate usually encodes a decision problem's solution, allowing us to verify its satisfiability. In the case of direct tokenisation, this certificate would be a vocabulary $\mathcal{S}$ which leads a dataset $\mathcal{D}$ to be compressed to at most $\delta$ symbols. Verifying this certificate simply requires computing the sum in Eq. (9), i.e.:

$$\sum_{\mathbf{c} \in \mathcal{D}} |\mathtt{tok}_\diamond[\mathcal{S}](\mathbf{c})| \qquad (10)$$

**Lemma 1.** *The direct tokenisation decision problem, as in Definition 3, is in NP.*

*Proof.* As noted above, whenever $|\mathcal{D}| \leq K$, each $\mathbf{c}_n \in \mathcal{D}$ can be included in the vocabulary $\mathcal{S}$ and fully compressed to a single symbol; we can thus verify the problem's satisfiability by simply checking that $\delta \geq |\mathcal{D}|$ as this is the

best reachable compression. Assuming $K$ to be bounded by $|\mathcal{D}|$—and therefore polynomial in the input—we have that the certificate $\mathcal{S}$ also has polynomial length. Given such a certificate $\mathcal{S}$, verifying it simply requires computing the sum in Eq. (10). In turn, computing this sum requires $|\mathcal{D}|$ calls to function $\mathtt{tok}_\diamond$. It follows that, if function $\mathtt{tok}_\diamond$ runs in polynomial time, then direct tokenisation is in NP. Luckily, this function can indeed be computed efficiently using Schmidt et al.'s (2024) PathPiece method, which runs in $O(|\mathbf{c}|^2)$ time. This is achieved by first converting $\mathbf{c}$ into a directed acyclic graph where nodes represent string positions $[0, 1, \ldots, |\mathbf{c}|]$ and where two nodes $t, t'$ are connected if there exists a subword $s \in \mathcal{S}$ for which $\mathbf{c}_{t:t'} \stackrel{\circ}{=} s$. E.g., for a string $\mathbf{c}$ and $\mathcal{S} = \Sigma \cup \{\mathbf{c}_{0:2}, \mathbf{c}_{1:t}\}$, we build a graph:

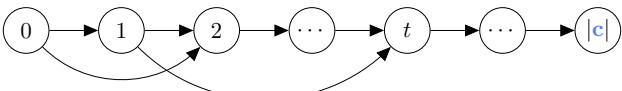

The shortest path from node $0$ to $|\mathbf{c}|$ in this graph then gives us $\mathtt{tok}_\diamond[\mathcal{S}](\mathbf{c})$. As the shortest path of a directed acyclic graph can be computed in $O(N + V)$ time, the time complexity of finding the shortest path in this directed acyclic graph is thus bounded by $O(|\mathbf{c}|^2)$. $\square$

## 5.2. Direct Tokenisation is NP-hard

We now use a reduction from max-2-SAT to prove the NP-hardness of direct tokenisation.

**Reduction 1.** *Let us have an instance of the* max-2-SAT *decision problem as in Definition 1. To reduce this instance to an instance of the direct tokenisation decision problem, as in Definition 3, we first define an alphabet* $\Sigma = \{\odot\} \cup \{x_j^\mathtt{T}, x_j^\mathtt{F}\}_{j=1}^J$*. We then construct three sets of strings:*

$$\mathcal{D}_1 = \{\odot x_j^\mathtt{T} \odot\}_{j=1}^J \cup \{\odot x_j^\mathtt{F} \odot\}_{j=1}^J \qquad (11a)$$
$$\mathcal{D}_2 = \{\odot x_j^\mathtt{T} \odot x_j^\mathtt{F} \odot\}_{j=1}^J \qquad (11b)$$
$$\mathcal{D}_3 = \{\odot L_i^1 \odot L_i^2 \odot\}_{i=1}^I \qquad (11c)$$

*In these strings* $L_i$ *is replaced by either character* $x_j^\mathtt{T}$ *or* $x_j^\mathtt{F}$*, depending on whether it represents* $X_j$ *or* $\neg X_j$*, respectively. We then construct our dataset* $\mathcal{D}$*, and choose* $K$ *and* $\delta$ *as:*

$$\mathcal{D} = \left( \bigcup_{\_=1}^f \mathcal{D}_1 \right) \cup \left( \bigcup_{\_=1}^{f'} \mathcal{D}_2 \right) \cup \mathcal{D}_3 \qquad (12a)$$

$$K = J, \quad \delta = (4f + 3f') J + 5I - 2\gamma \qquad (12b)$$

*where we set* $f' \stackrel{\mathrm{def}}{=} 2I + 1$ *and* $f \stackrel{\mathrm{def}}{=} 4f'J + 4I + 1$.

We write $\mathrm{R1}(\mathcal{X}, \mathcal{C}, \gamma)$ to represent a function which, given an instance of max-2-SAT, returns an instance of the tokenisation problem given by our reduction (i.e., $\mathcal{D}, K, \delta$).

For our reduction to be correct, we must have that:

$$M2S(\mathcal{X}, \mathcal{C}, \gamma) \iff Tok_\phi(R1(\mathcal{X}, \mathcal{C}, \gamma)) \qquad (13)$$

meaning that a `max-2-SAT` instance is satisfiable if and only if its reduced direct tokenisation instance is as well. We now set out to prove this. We start by proving the forward direction of this iff clause.

**Lemma 2.** *If a* `max-2-SAT` *instance is satisfiable, then the direct tokenisation instance output by Reduction 1 is also satisfiable. Formally:*

$$M2S(\mathcal{X}, \mathcal{C}, \gamma) \implies Tok_\phi(R1(\mathcal{X}, \mathcal{C}, \gamma)) \qquad (14)$$

*Proof sketch.* See a formal proof in App. A. Our proof works by first fixing a satisfying solution to `max-2-SAT` with values $x_j^\star$. Given this solution, for each variable, we add to our vocabulary $\mathcal{S}$ a subword $\odot x_j^{\mathsf{T}} \odot$ if $x_j^\star$ is true, or $\odot x_j^{\mathsf{F}} \odot$ if $x_j^\star$ is false. Given these subwords, strings in $\mathcal{D}_1$ and $\mathcal{D}_2$ occupy a total length of $(4f + 3f') J$. Further, since at least $\gamma$ of the `max-2-SAT` clauses are satisfied by $x_j^\star$, the strings in $\mathcal{D}_3$ will occupy a total length smaller or equal to $5I - 2\gamma$. This solution to the tokenisation problem thus gives us a total length which is smaller or equal to $\delta = (4f + 3f') J + 5I - 2\gamma$. □

Now, we are left with proving the backward direction of the iff clause in Eq. (13). We do so in the following lemma.

**Lemma 3.** *If the direct tokenisation instance output by Reduction 1 is satisfiable, the* `max-2-SAT` *instance which generated it is as well. Formally:*

$$Tok_\phi(R1(\mathcal{X}, \mathcal{C}, \gamma)) \implies M2S(\mathcal{X}, \mathcal{C}, \gamma) \qquad (15)$$

*Proof sketch.* See a formal proof in App. B. Our proof works in three steps. First, we show that any satisfying solution must only have subwords of the form $\odot x_j^{\mathsf{T}} \odot$ or $\odot x_j^{\mathsf{F}} \odot$, since this is required to compress strings in $\mathcal{D}_1$ to at most $4fJ$ symbols. Second, we show that any satisfying solution must only have either subword $\odot x_j^{\mathsf{T}} \odot$ or $\odot x_j^{\mathsf{F}} \odot$ for any variable $X_j$; this is required to compress strings in $\mathcal{D}_2$ to at most $3f'J$ symbols. Finally, we show that if a tokeniser compresses strings in $\mathcal{D}_3$ to $5I - 2\gamma$, then there is an assignment $\chi$ which satisfies at least $\gamma$ of the original `max-2-SAT` problem. □

Given both lemmas above, we can now trivially prove that direct tokenisation is NP-hard.

**Lemma 4.** *The direct tokenisation decision problem, as in Definition 3, is NP-hard.*

*Proof.* First, it is easy to see that Reduction 1 runs in polynomial time. Second, `max-2-SAT` is an NP-hard problem (Garey et al., 1974). This lemma then follows trivially from

Lemmas 2 and 3, which together show that an instance of the tokenisation problem generated through Reduction 1 is satisfiable if and only if the `max-2-SAT` instance used to produce it is also satisfiable. □

## 6. Finding Optimal Bottom-up Tokenisers

We now shift our attention to bottom-up tokenisation. We define both its optimisation and decision problems, and then prove its NP-completeness. We start with defining the optimisation problem.

**Definition 4.** *Given a dataset $\mathcal{D}$ and a vocabulary size $K$, the **bottom-up tokenisation optimisation problem** is to find a merge sequence $\mathbf{m}_{\mathrm{opt}} \in \mathcal{M}^*$ which maximally compresses $\mathcal{D}$:*

$$\mathbf{m}_{\mathrm{opt}} = \arg\min_{\mathbf{m} \in \mathcal{M}^*} \sum_{\mathbf{c} \in \mathcal{D}} |tok_\uparrow[\mathbf{m}](\mathbf{c})| \qquad (16)$$
$$\text{s.t. } |\mathbf{m}| = K$$

As can be seen, this optimisation problem is similar to the direct tokenisation problem, albeit its target is to find a merge sequence instead of a vocabulary. We similarly define a decision problem.

**Definition 5.** *Given a dataset $\mathcal{D}$ and a vocabulary size $K$, the **bottom-up tokenisation decision problem** requires deciding whether there exists a merge sequence $\mathbf{m} \in \mathcal{M}^*$ which compresses $\mathcal{D}$ to at most $\delta$ symbols:*

$$\delta \geq \min_{\mathbf{m} \in \mathcal{M}^*} \sum_{\mathbf{c} \in \mathcal{D}} |tok_\uparrow[\mathbf{m}](\mathbf{c})| \qquad (17)$$
$$\text{s.t. } |\mathbf{m}| = K$$

We write $Tok_\uparrow(\mathcal{D}, K, \delta)$ for a function which returns T if a bottom-up tokenisation decision problem with those inputs is satisfiable, and F otherwise. We spend the rest of this section showing that bottom-up tokenisers are NP-complete.

**Theorem 2.** *The bottom-up tokenisation decision problem, as in Definition 5, is NP-complete.*

*Proof.* We prove this in two steps below. We first prove that this problem is in NP, in §6.1. We then prove that this problem is NP-hard, in §6.2. □

### 6.1. Bottom-up Tokenisation is in NP

We can verify this using a solution, the merge sequence $\mathbf{m} \in \mathcal{M}^*$, as a certificate. By showing that this certificate has polynomial length and that it can be verified in polynomial time, we prove this problem is in NP. To verify this certificate, we simply need to compute the sum in Eq. (17), i.e.:

$$\sum_{\mathbf{c} \in \mathcal{D}} |tok_\uparrow[\mathbf{m}](\mathbf{c})| \qquad (18)$$

which we show now can be done efficiently.

**Lemma 5.** *The bottom-up tokenisation decision problem, as in Definition 5, is in NP.*

*Proof.* First, if $K$ is larger than the total number of characters in $\mathcal{D}$, i.e., $\sum_{\mathbf{c} \in \mathcal{D}} |\mathbf{c}|$, then this dataset can be compressed to $|\mathcal{D}|$ by merging each string down to a single symbol; further, compressing $\mathcal{D}$ more than that is not possible independently of $K$. Verifying the satisfiability of such an instance of the tokenisation problem is thus trivial, only requiring checking if $\delta \geq |\mathcal{D}|$. Second, if $K$ is bounded by $\sum_{\mathbf{c} \in \mathcal{D}} |\mathbf{c}|$—and therefore polynomial in the input—the certificate $\mathbf{m}$ has polynomial length. Given such a certificate $\mathbf{m}$, verifying it then simply requires computing the sum in Eq. (18). In turn, computing this sum requires $|\mathcal{D}|$ calls to function $\mathtt{tok}_\uparrow$. It follows that, if function $\mathtt{tok}_\uparrow$ runs in polynomial time, then bottom-up tokenisation is in NP. The computation of $\mathtt{tok}_\uparrow$, can be done in polynomial time following the structure described in §3.2. For each $m = \langle s_1, s_2 \rangle$ in $\mathbf{m}$, scan the current $\mathbf{c}$ and replace each occurrence of $s_1, s_2$ by $s'$. This takes time $\mathcal{O}(|\mathbf{c}|)$ for each merge. Afterwards, the resulting string can be compared against the desired size. We obtain a total runtime of $O(|\mathcal{D}||\mathbf{c}||\mathbf{m}|)$. $\square$

## 6.2. Bottom-up Tokenisation is NP-hard

As before, we use a reduction from $\mathtt{max\text{-}2\text{-}SAT}$ to prove bottom-up tokenisation's NP-hardness.

**Reduction 2.** *Let us have an instance of the $\mathtt{max\text{-}2\text{-}SAT}$ decision problem as in Definition 1. To reduce this instance to an instance of the bottom-up tokenisation decision problem, as in Definition 5, we first define an alphabet $\Sigma = \{\odot, \otimes\} \cup \{x_j^\mathtt{T}, x_j^\mathtt{F}\}_{j=1}^J$. We then construct five sets of strings:*

$$\mathcal{D}_1 = \{\odot x_j^\mathtt{T}\}_{j=1}^J \cup \{x_j^\mathtt{F}\odot\}_{j=1}^J \cup \{x_j^\mathtt{T}\odot\}_{j=1}^J \tag{19}$$
$$\cup \{\odot x_j^\mathtt{F}\}_{j=1}^J \cup \{x_j^\mathtt{T}\otimes\}_{j=1}^J \cup \{\otimes x_j^\mathtt{F}\}_{j=1}^J$$
$$\mathcal{D}_2 = \{\odot x_j^\mathtt{T}\odot\}_{j=1}^J \cup \{\odot x_j^\mathtt{F}\odot\}_{j=1}^J$$
$$\cup \{\odot x_j^\mathtt{T}\otimes\}_{j=1}^J \cup \{\otimes x_j^\mathtt{F}\odot\}_{j=1}^J$$
$$\mathcal{D}_3 = \{\odot x_j^\mathtt{T}\odot x_j^\mathtt{F}\odot\}_{j=1}^J \cup \{\otimes x_j^\mathtt{F}\odot x_j^\mathtt{T}\otimes\}_{j=1}^J$$
$$\mathcal{D}_4 = \{\odot x_j^\mathtt{F}\odot x_j^\mathtt{T}\otimes\}_{j=1}^J \cup \{\otimes x_j^\mathtt{F}\odot x_j^\mathtt{T}\odot\}_{j=1}^J$$
$$\mathcal{D}_5 = \begin{Bmatrix} \odot x_j^\mathtt{T}\odot x_{j'}^\mathtt{F}\odot & \text{if } L_i^1 = X_j & \text{and } L_i^2 = \neg X_{j'} \\ \odot x_{j'}^\mathtt{T}\odot x_j^\mathtt{F}\odot & \text{if } L_i^1 = \neg X_j & \text{and } L_i^2 = X_{j'} \\ \otimes x_j^\mathtt{F}\odot x_{j'}^\mathtt{F}\odot & \text{if } L_i^1 = \neg X_j & \text{and } L_i^2 = \neg X_{j'} \\ \odot x_j^\mathtt{T}\odot x_{j'}^\mathtt{T}\otimes & \text{if } L_i^1 = X_j & \text{and } L_i^2 = X_{j'} \end{Bmatrix}_{i=1}^I$$

*We then construct our dataset $\mathcal{D}$, and choose $K$ and $\delta$ as:*

$$\mathcal{D} = \overset{f}{\bigcup}\mathcal{D}_1 \cup \overset{f'}{\bigcup}\mathcal{D}_2 \cup \overset{f''}{\bigcup}\mathcal{D}_3 \cup \overset{f'''}{\bigcup}\mathcal{D}_4 \cup \mathcal{D}_5 \tag{20}$$
$$K = 8J, \ \delta = (6f + 6f' + 4f'' + 4f''')\,J + 3\,I - \gamma$$

*where we set:*

$$f''' \overset{\text{def}}{=} 5I, \quad f'' \overset{\text{def}}{=} 10f'''J + 5I \tag{21a}$$
$$f' \overset{\text{def}}{=} (10f'' + 10f''')\,J + 5I \tag{21b}$$
$$f \overset{\text{def}}{=} (12f' + 10f'' + 10f''')\,J + 5I \tag{21c}$$

As before, we write $\mathrm{R2}(\mathcal{X}, \mathcal{C}, \gamma)$ for a function which, given an instance of the $\mathtt{max\text{-}2\text{-}SAT}$ problem, returns an instance of the bottom-up tokenisation problem. For our reduction to be correct, we must have that:

$$\mathrm{M2S}(\mathcal{X}, \mathcal{C}, \gamma) \iff \mathrm{Tok}_\uparrow(\mathrm{R2}(\mathcal{X}, \mathcal{C}, \gamma)) \tag{22}$$

We follow the same proof strategies as before, starting by proving the forward direction of this iff statement.

**Lemma 6.** *If a $\mathtt{max\text{-}2\text{-}SAT}$ instance is satisfiable, then the bottom-up tokenisation instance output by Reduction 2 is also satisfiable. Formally:*

$$\mathrm{M2S}(\mathcal{X}, \mathcal{C}, \gamma) \implies \mathrm{Tok}_\uparrow(\mathrm{R2}(\mathcal{X}, \mathcal{C}, \gamma)) \tag{23}$$

*Proof sketch.* See a formal proof in App. C. Without loss of generality, let a satisfying solution to $\mathtt{max\text{-}2\text{-}SAT}$ have values $x_j^\star$. Our proof works by first defining the three following lists of merges, which must be included in any satisfying solution to this tokenisation problem:

$$\mathbf{m}_1 = \bigcirc_{j=1}^J [\langle \otimes, x_j^\mathtt{F}\rangle, \langle x_j^\mathtt{T}, \otimes\rangle] \tag{24a}$$
$$\mathbf{m}_3 = \bigcirc_{j=1}^J [\langle x_j^\mathtt{F}, \odot\rangle, \langle \odot, x_j^\mathtt{T}\rangle] \tag{24b}$$
$$\mathbf{m}_5 = \bigcirc_{j=1}^J [\langle \odot, x_j^\mathtt{F}\rangle, \langle x_j^\mathtt{T}, \odot\rangle] \tag{24c}$$

We then construct two other lists of merges, which depend on the satisfying assignments to $\mathtt{max\text{-}2\text{-}SAT}$:

$$\mathbf{m}_2 = \bigcirc_{j=1}^J \begin{bmatrix} \langle \odot, x_j^\mathtt{T}\otimes\rangle & \text{if } x_j^\star = \mathtt{T} \\ \langle \otimes x_j^\mathtt{F}, \odot\rangle & \text{else} \end{bmatrix} \tag{25a}$$
$$\mathbf{m}_4 = \bigcirc_{j=1}^J \begin{bmatrix} \langle \odot x_j^\mathtt{T}, \odot\rangle & \text{if } x_j^\star = \mathtt{T} \\ \langle \odot, x_j^\mathtt{F}\odot\rangle & \text{else} \end{bmatrix} \tag{25b}$$

Finally, we create a merge sequence by concatenating these lists in order:

$$\mathbf{m} = \mathbf{m}_1 \circ \mathbf{m}_2 \circ \mathbf{m}_3 \circ \mathbf{m}_4 \circ \mathbf{m}_5 \tag{26}$$

Note that we have exactly $K = 8J$ merges in this list. Given this merge sequence, it is easy to verify that strings in $\mathcal{D}_1$ to $\mathcal{D}_4$ will use exactly $(6f + 6f' + 4f'' + 4f''')\,J$ symbols after being tokenised. Further, since at least $\gamma$ of the $\mathtt{max\text{-}2\text{-}SAT}$'s clauses are satisfied by $x_j^\star$, the strings in $\mathcal{D}_5$ will occupy a total length smaller or equal to $3\,I - \gamma$. This solution to the tokenisation problem thus gives us a tokeniser which will compress $\mathcal{D}$ to at most $\delta = (6f + 6f' + 4f'' + 4f''')\,J + 3\,I - \gamma$. $\square$

We now prove the backward direction of the iff clause in Eq. (22).

**Lemma 7.** *If the bottom-up tokenisation instance output by Reduction 2 is satisfiable, the* max-2-SAT *instance which generated it is as well. Formally:*

$$\text{Tok}_\uparrow(\text{R2}(\mathcal{X}, \mathcal{C}, \gamma)) \implies \text{M2S}(\mathcal{X}, \mathcal{C}, \gamma) \qquad (27)$$

*Proof sketch.* See a formal proof in App. D. Our proof works in five steps. First, we show that all satisfying solutions must include merges $\mathbf{m}_1$, $\mathbf{m}_3$, and $\mathbf{m}_5$ from Eq. (24), since this is required to compress strings in $\mathcal{D}_1$ to at most $6fJ$ symbols. Second, we show the other merges of any satisfying solution must be of the form:

$$\mathbf{m}_j^{\circledcirc} = \left\{ \begin{array}{l} \langle \circledcirc x_j^\mathsf{T}, \circledcirc \rangle, \langle \circledcirc, x_j^\mathsf{F} \circledcirc \rangle \\ \langle \circledcirc, x_j^\mathsf{T} \circledcirc \rangle, \langle \circledcirc x_j^\mathsf{F} \circledcirc, \circledcirc \rangle \end{array} \right\} \qquad (28a)$$

$$\mathbf{m}_j^{\otimes} = \left\{ \begin{array}{l} \langle \circledcirc, x_j^\mathsf{T} \otimes \rangle, \langle \otimes x_j^\mathsf{F}, \circledcirc \rangle \\ \langle \circledcirc x_j^\mathsf{T}, \otimes \rangle, \langle \otimes, x_j^\mathsf{F} \circledcirc \rangle \end{array} \right\} \qquad (28b)$$

this is required to compress strings in $\mathcal{D}_2$ to at most $6f'J$ symbols. Third, we show that any satisfying solution will have at least one merge of each set $\mathbf{m}_j^{\circledcirc}$ and one of each set $\mathbf{m}_j^{\otimes}$; this is required to compress strings in $\mathcal{D}_3$ to at most $4f''J$ symbols. Fourth, we show that any satisfying solution will have—for each $j \in \{1, ..., J\}$—both its merges in sets $\mathbf{m}_j^{\circledcirc}$ and $\mathbf{m}_j^{\otimes}$ containing character $x_j^\mathsf{T}$ or character $x_j^\mathsf{F}$; this is required to compress strings in $\mathcal{D}_4$ to at most $4f'''J$ symbols. Finally, we show that if a tokeniser compresses strings in $\mathcal{D}_5$ to $3I - \gamma$, then there is an assignment $\chi$ which satisfies at least $\gamma$ of the original max-2-SAT problem. $\square$

Finally, given both lemmas above, we can now prove that bottom-up tokenisation is NP-hard.

**Lemma 8.** *The bottom-up tokenisation decision problem, as in Definition 5, is NP-hard.*

*Proof.* First, it is easy to see that Reduction 2 runs in polynomial time. Second, max-2-SAT is an NP-hard problem (Garey et al., 1974). This lemma then follows trivially from Lemmas 6 and 7. $\square$

### 6.3. Other Definitions of Tokenisation

We now expand our discussion to consider variations of the above tokenisation problems.

**Deduped Datasets.** Our definitions of both direct and bottom-up tokenisation allow datasets $\mathcal{D}$ to include repeated entries. It is common, however, to deduplicate datasets in NLP—thus removing repeated entries. A small change to both our reductions is enough to adapt it to this deduplicated dataset case: simply append each string in the repeated datasets (either $\mathcal{D}_1$ and $\mathcal{D}_2$ in Reduction 1 or $\mathcal{D}_1$ to $\mathcal{D}_4$ in

Reduction 2) with a unique character $\{a_y\}_{y=1}^\infty$ and increase the target compression size $\delta$ accordingly (by $f + f'$ or $f + f' + f'' + f'''$, respectively). These new characters will never be included in optimal tokenisers' solutions, and thus the previous proofs hold, with the difference that each dataset will require extra symbols once compressed.

**A Single Long String.** In the previous sections, we considered tokenisers trained on a dataset $\mathcal{D}$. Work on compression, however, usually considers a single long string $\mathbf{c}$ as its input. It is easy to see that direct tokenisation is not an NP-complete problem if its input is a single long string; including this string in vocabulary $\mathcal{S}$ already achieves optimal compression. Bottom-up tokenisation, however, is still NP-complete even when given a single string as input. As before, this can be shown with a similar strategy to Reduction 2, but where we first append each string in dataset $\mathcal{D}$ with a unique character $\{a_y\}_{y=1}^\infty$ and then concatenate all these strings. As in the deduped case above, characters $a_y$ will never be merged by any optimal tokeniser; they will thus serve as virtual string delimiters and will not affect our proofs beyond an increase to the target compression size $\delta$.

**A Hybrid Approach.** Finally, the last variant we consider is a hybrid between direct and bottom-up tokenisation, where we find a merge sequence $\mathbf{m}$ which—when we extract a vocabulary from it as $\mathcal{S} = \Sigma \cup \{s_1 \circ s_2 \mid \langle s_1, s_2 \rangle \in \mathbf{m}\}$— optimally compresses a dataset $\mathcal{D}$ using the direct tokenisation function in Eq. (4). We can easily prove the NP-hardness of this tokenisation variant by relying on Reduction 2; as our proof in Lemma 8 did not make use of the order of merges in $\mathbf{m}$, only of the subwords composed by it, this lemma's proof strategy can be similarly applied to this hybrid variant.

## 7. Tokenisation and Compression

The variants of tokenisation that we consider here—with compression as their objective function—are closely related to the field of dictionary compression. In both fields, we wish to reduce the size of an input ($\mathbf{c}$ or $\mathcal{D}$) by exploiting repetitive elements. In fact, the most popular tokenisation algorithm to date, BPE, was originally proposed as a compression algorithm (Gage, 1994) and has only somewhat recently been ported into NLP to find tokenisers (by Sennrich et al., 2016).

Not surprisingly, prior work has also considered, from a theoretical perspective, the compression tokenisers achieve. Zouhar et al. (2023b), for instance, analyse bottom-up tokenisation and prove an approximation bound on the compression achieved by the tokenisers found using BPE. More recently, Kozma & Voderholzer (2024) also analyses bottom-up tokenisation, proving a tighter bound on this compression

achieved by BPE.

A popular dictionary compression method, the **straight-line program** (SLP; Kieffer & Yang, 2000; Charikar et al., 2005), can be used to illustrate the similarities and differences between tokenisers and compressors.[11] Given a string **c**, an SLP describes a context-free grammar from which **c** can be uniquely derived. Formally, an SLP in Chomsky normal form (CNF) is a set of rules of form $X \to a$ or $X \to AB$, where $X, A, B$ are called nonterminals and $a$ is a terminal.[12] Starting from a special nonterminal $S$, applying these rules exhaustively—until only terminals are left—produces exactly the desired string **c**. Notably, given a string **c**, it is NP-complete to find the smallest SLP which generates it (Charikar et al., 2005).

On the one hand, SLPs in CNF are closely linked to bottom-up tokenisation; each of its rules expands to two nonterminals, and thus corresponds to a merge. However, while SLPs must find the minimum number of merges (or rules) to fully compress a string into a single symbol, bottom-up tokenisers must maximally compress the string given a fixed number of merges. On the other hand, SLPs which are not in CNF (that is, for which other context-free production rules are allowed, as long as the decoding stays unique) are closely linked to direct tokenisation. In this case, a direct tokeniser could be converted into an SLP with depth two; this grammar has a start rule $S \to \mathbf{s}$, and a rule from each subword to its characters $s \to \mathbf{c}$. Again, while SLPs must find a minimal grammar representing the string, direct tokenisers must minimise the size of rule $S \to \mathbf{s}$ given a fixed number of rules $s \to \mathbf{c}$.

The paragraphs above highlights two important differences between tokenisers and compressors. First, tokenisers aim to reduce only the size of the resulting tokenised text (i.e., $|\mathbf{s}|$), whereas compressors also consider the size of the compression information (e.g., considering the size required to store $S$, which would be $\sum_{s \in \mathcal{S}} |\mathtt{detok}(s)|$). This is because tokenisers must create shorter inputs for NLP algorithms, while compressors must make information compact. Second, tokenisers and compressors have different optimisation parameters. Compression algorithms always compress a string to the best extent possible (e.g., for SLPs, until a single nonterminal is reached), whereas tokenisation algorithms are given a maximum vocabulary size (i.e., $K$) and find tokenisers which only compress their input as much as possible until this limit is reached.

---

[11]See Lohrey (2012) for an overview of straight-line programs, and Kempa & Prezza (2018); Kociumaka et al. (2023) for a more detailed overview of compression in general.

[12]Although not originally defined that way, SLP's grammars are typically assumed to be in CNF, for simplicity. This does not make a big difference for compression, but will be important for our purposes.

## 8. Conclusion

In this work, we proved the NP-completeness of two variants of tokenisation. These results underline that finding optimal tokenisers most likely will remain a difficult quest and that research should focus on approximate algorithms instead. Regarding those, there is potential both in improving the analysis of currently used algorithms, such as BPE, as well as in designing new ones.

## Limitations

While we prove the NP-completeness of multiple variants of the tokenisation problem—which is an important part of modern language modelling pipelines—we must note a few limitations in our work. First, we only prove NP-completeness of tokenisation with compression as its objective. This is a popular objective function, frequently used to judge the quality of tokenisers (e.g., Liu et al., 2025); however, it is not perfectly correlated with downstream language modelling performance, as discussed in §2.[13] Investigating the complexity of tokenisation under other objective functions is important. Second, our proofs do not assume a fixed alphabet size, so for fixed alphabets tokenisation might not be NP-complete. Tokenisers are frequently run at the byte level, for which specialised, more efficient algorithms might exist. Finally, while we investigated the complexity of the tokenisation problem for two types of tokenisation functions, similar results for other variants (with other tokenisation functions) remain open; we believe this would be exciting future work.

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

# A. Proof of Lemma 2

**Lemma 2.** *If a* max-2-SAT *instance is satisfiable, then the direct tokenisation instance output by Reduction 1 is also satisfiable. Formally:*

$$\mathrm{M2S}(\mathcal{X}, \mathcal{C}, \gamma) \implies \mathrm{Tok}_{\Diamond}(\mathrm{R1}(\mathcal{X}, \mathcal{C}, \gamma)) \tag{14}$$

*Proof.* First, note that if $\mathrm{M2S}(\mathcal{X}, \mathcal{C}, \gamma)$, then we have that Eq. (7) holds: $\gamma \leq \max_{\chi \in \{\mathtt{F},\mathtt{T}\}^J} \sum_{i=1}^{I} \mathbb{1}\{L_i^1 \vee L_i^2\}$. Now, without loss of generality, let a satisfying solution have values $x^\star$. In this case, for each variable $X_j$, we construct token $\odot x_j^\mathtt{T} \odot$ if $x_j^\star$ is true, or $\odot x_j^\mathtt{F} \odot$ if $x_j^\star$ is false. This gives us a total of $J$ new tokens, so satisfies the $|\mathcal{S}| = |\Sigma| + K$ condition. Now we just need to count the symbols output by this solution to see if Eq. (9) is satisfied, since any given tokenisation $\mathtt{tok}(\cdot, \mathcal{S})$ will provide an upper bound on the optimal tokenisation in terms of compression:

$$\sum_{\mathbf{c} \in \mathcal{D}} |\mathtt{tok}_{\Diamond}[\mathcal{S}](\mathbf{c})| \geq \min_{\mathcal{S}' \subset \Sigma^+} \sum_{\mathbf{c} \in \mathcal{D}} |\mathtt{tok}_{\Diamond}[\mathcal{S}'](\mathbf{c})| \tag{29}$$
$$\text{s.t. } |\mathcal{S}'| = |\Sigma| + K$$

For each pair of strings $\odot x_j^\mathtt{T} \odot$ and $\odot x_j^\mathtt{F} \odot$ in $\mathcal{D}_1$, one is compressed into a single subword while the other is kept as originally—using 3 symbols. We thus have that the strings in $\mathcal{D}_1$ will occupy a total of $(1+3)J$ characters, and:

$$\sum_{\mathbf{c} \in (\bigcup_{-=1}^{f} \mathcal{D}_1)} |\mathtt{tok}_{\Diamond}[\mathcal{S}](\mathbf{c})| = 4fJ \tag{30}$$

Similarly, for each string in $\mathcal{D}_2$ of form $\odot x_j^\mathtt{T} \odot x_j^\mathtt{F} \odot$, we have that either token $\odot x_j^\mathtt{T} \odot$ or $\odot x_j^\mathtt{F} \odot$ exists. So each of these strings is compressed from 5 into 3 symbols. We thus have:

$$\sum_{\mathbf{c} \in (\bigcup_{-=1}^{f'} \mathcal{D}_2)} |\mathtt{tok}_{\Diamond}[\mathcal{S}](\mathbf{c})| = 3f'J \tag{31}$$

Finally, we have strings in $\mathcal{D}_3$ of form $\odot L_i^1 \odot L_i^2 \odot$. These strings will be compressed into 3 symbols if $\odot L_i^1 \odot$ or $\odot L_i^2 \odot$ (or both) exist, and kept with 5 symbols otherwise. We thus have:

$$\sum_{\mathbf{c} \in \mathcal{D}_3} |\mathtt{tok}_{\Diamond}[\mathcal{S}](\mathbf{c})| = \sum_{i=1}^{I} \left( 5 - 2 \, \mathbb{1} \left\{ \begin{array}{c} \odot L_i^1 \odot \in \mathcal{S} \\ \text{or} \\ \odot L_i^2 \odot \in \mathcal{S} \end{array} \right\} \right) \tag{32a}$$

$$= 5I - 2 \sum_{i=1}^{I} \mathbb{1} \left\{ \begin{array}{c} \odot x_j^\mathtt{T} \odot \in \mathcal{S} \text{ and } L_i^1 = X_j \\ \text{or} \\ \odot x_j^\mathtt{F} \odot \in \mathcal{S} \text{ and } L_i^1 = \neg X_j \\ \text{or} \\ \odot x_{j'}^\mathtt{T} \odot \in \mathcal{S} \text{ and } L_i^2 = X_{j'} \\ \text{or} \\ \odot x_{j'}^\mathtt{F} \odot \in \mathcal{S} \text{ and } L_i^2 = \neg X_{j'} \end{array} \right\} \tag{32b}$$

$$= 5I - 2 \sum_{i=1}^{I} \mathbb{1}_{\chi^\star} \{ L_i^1 \vee L_i^2 \} \tag{32c}$$

$$\leq 5I - 2\gamma \tag{32d}$$

where, by construction, we have that a subword $\odot L_i \odot \in \mathcal{S}$ if and only if its associated variable ($X_j$ or $\neg X_j$) is true. Summing together the lengths in Eqs. (30) to (32), we get that

$$\sum_{\mathbf{c} \in \mathcal{D}} |\mathtt{tok}_{\Diamond}[\mathcal{S}](\mathbf{c})| \leq \delta = (4f + 3f') J + 5I - 2\gamma \tag{33}$$

which concludes the proof. $\qquad\square$

# B. Proof of Lemma 3

**Lemma 3.** *If the direct tokenisation instance output by Reduction 1 is satisfiable, the* max-2-SAT *instance which generated it is as well. Formally:*

$$\text{Tok}_\phi(\text{R1}(\mathcal{X}, \mathcal{C}, \gamma)) \implies \text{M2S}(\mathcal{X}, \mathcal{C}, \gamma) \tag{15}$$

*Proof.* First, note that the dataset $\mathcal{D}$ output by Reduction 1 has a total number of characters:

$$\sum_{\mathbf{c} \in \mathcal{D}} |\mathbf{c}| = (6f + 5f')J + 5I \tag{34}$$

Further, let:

$$\texttt{toklen}(\mathcal{D}, \mathcal{S}) \overset{\text{def}}{=} \sum_{\mathbf{c} \in \mathcal{D}} |\texttt{tok}_\phi[\mathcal{S}](\mathbf{c})|, \qquad \mathcal{S}_0 = \Sigma \cup \bigcup_{j=1}^{J} \{\odot x_j^\text{T} \odot, \odot x_j^\text{F} \odot\} \tag{35}$$

The maximum number of symbols in this dataset after compression is set to $\delta = (4f + 3f')\,J + 5\,I - 2\gamma$. This means that, to satisfy this objective, there must exist a vocabulary whose tokeniser compresses the text by at least $(2f + 2f')\,J + 2\gamma$ symbols. We now prove this lemma in three steps: ① we show that any solution which compresses the text by at least $2fJ$ symbols must only have nontrivial subwords[14] of the form $\odot x_j^\text{T} \odot$ or $\odot x_j^\text{F} \odot$; ② we show that any solution which compresses the text by at least $(2f + 2f')J$ symbols must only have either subword $\odot x_j^\text{T} \odot$ or $\odot x_j^\text{F} \odot$ for any variable $X_j$; ③ we show that any solution which compresses the text by at least $(2f + 2f')J + 2\gamma$ symbols must be produced by a max-2-SAT instance which has at least $\gamma$ clauses that are simultaneously satisfiable. $\qquad \square$

**LemmaProofStep 1.** (Step ①). *Any solution which compresses the text by at least $2fJ$ symbols must only have nontrivial subwords of the form $\odot x_j^\text{T} \odot$ or $\odot x_j^\text{F} \odot$, i.e.,:*

$$\left( \texttt{toklen}(\mathcal{D}, \mathcal{S}) \leq \underbrace{(4f + 5f')J + 5I}_{\sum_{\mathbf{c} \in \mathcal{D}} |\mathbf{c}| - 2fJ} \right) \implies \mathcal{S} \subset \mathcal{S}_0 \tag{36}$$

*Proof.* First, given a solution with $\mathcal{S} \subset \mathcal{S}_0$, each subword $s \in \mathcal{S}_{\backslash \Sigma}$, where $\mathcal{S}_{\backslash \Sigma} \overset{\text{def}}{=} \mathcal{S} \setminus \Sigma$, will replace at least $f$ strings in $\mathcal{D}_1$—i.e., with form $\odot x_j^\text{T} \odot$ or $\odot x_j^\text{F} \odot$—for a single subword, thus saving $2f$ characters. Since we have $|\mathcal{S}_{\backslash \Sigma}| = K = J$ tokens, we save exactly $2fJ$ symbols in $\mathcal{D}_1$:

$$\mathcal{S} \subset \mathcal{S}_0 \implies \left( \texttt{toklen}(\mathcal{D}_1, \mathcal{S}') = \underbrace{4fJ}_{\sum_{\mathbf{c} \in \mathcal{D}_1} |\mathbf{c}| - 2fJ} \right) \tag{37}$$

Note now that any solution $\mathcal{S}'$ for which $\mathcal{S}' \not\subset \mathcal{S}_0$ has at least one nontrivial subword which is not of the form $\odot x_j^\text{T} \odot$ or $\odot x_j^\text{F} \odot$; this subword $s \notin \mathcal{S}_0$ will thus not compress strings in $\mathcal{D}_1$ by $2f$ symbols, but by at most $f$ symbols:

$$\mathcal{S}' \not\subset \mathcal{S}_0 \implies \left( \texttt{toklen}(\mathcal{D}_1, \mathcal{S}') \geq \underbrace{4f(J-1) + 5f}_{\sum_{\mathbf{c} \in \mathcal{D}_1} |\mathbf{c}| - 2fJ + f} \right) \tag{38}$$

Even if this new subword were able to fully compress strings in $\mathcal{D}_2$ and $\mathcal{D}_3$ to a single symbol each, it would reach a compression of at most $4f'J + 4I$. Since by design $f = 4f'J + 4I + 1$, we get that:

$$\mathcal{S}' \not\subset \mathcal{S}_0 \implies \left( \texttt{toklen}(\mathcal{D}, \mathcal{S}') \geq 4fJ + f + f'J + I > (4f + 5f')J + 5I \right) \tag{39}$$

which concludes this step of the proof. $\qquad \square$

---

[14]We define nontrivial subwords as subwords with more than one character. Remember that by definition $\Sigma \subseteq \mathcal{S}$, so all characters are always included in tokenisers' vocabularies. Also note that $|\mathcal{S}| = |\Sigma| + K$, so those trivial subwords are not counted towards vocabulary size $K$.

**LemmaProofStep 2.** (Step ②). *Any solution which compresses the text by at least $(2f + 2f')J$ symbols must only have either subword $\odot x_j^{\mathsf{T}}\odot$ or $\odot x_j^{\mathsf{F}}\odot$ for any variable $X_j$, i.e.,:*

$$\left(\texttt{toklen}(\mathcal{D},\mathcal{S}) \leq \underbrace{(4f + 3f')J + 5I}_{\sum_{\mathbf{c}\in\mathcal{D}}|\mathbf{c}| - (2f+2f')J}\right) \implies \forall_{j\in\{1,\dots,J\}}\ |\mathcal{S}\cap\{\odot x_j^{\mathsf{T}}\odot, \odot x_j^{\mathsf{F}}\odot\}| = 1 \tag{40}$$

*Proof.* In this step of the proof, we show that satisfying solutions must have one and only one of subwords $\odot x_j^{\mathsf{T}}\odot$ and $\odot x_j^{\mathsf{F}}\odot$ for any variable $X_j$. As before, it's easy to see that a solution of the form described achieves at least $(2f + 2f')J$ symbol compression. Each subword of form $\odot x_j^{\mathsf{T}}\odot$ or $\odot x_j^{\mathsf{F}}\odot$ saves exactly $2f$ characters in the strings in $\mathcal{D}_1$. Further, because we always have either subword $\odot x_j^{\mathsf{T}}\odot$ or $\odot x_j^{\mathsf{F}}\odot$ for each value of $j$, we also get $2f'$ compression in the strings in $\mathcal{D}_2$:

$$\forall_{j\in\{1,\dots,J\}}\ |\mathcal{S}\cap\{\odot x_j^{\mathsf{T}}\odot, \odot x_j^{\mathsf{F}}\odot\}| = 1 \tag{41}$$
$$\implies \left(\texttt{toklen}(\mathcal{D}_1,\mathcal{S}) = \underbrace{4fJ}_{\sum_{\mathbf{c}\in\mathcal{D}_1}|\mathbf{c}| - 2fJ}\right) \text{ and } \left(\texttt{toklen}(\mathcal{D}_2,\mathcal{S}) = \underbrace{3f'J}_{\sum_{\mathbf{c}\in\mathcal{D}_2}|\mathbf{c}| - 2f'J}\right)$$

Now note that this is not true if both $\odot x_j^{\mathsf{T}}\odot$ and $\odot x_j^{\mathsf{F}}\odot$ exist for a single $j$; in this case, only one of the tokens can be applied to $\odot x_j^{\mathsf{T}}\odot x_j^{\mathsf{F}}\odot$, and thus both tokens together lead to a benefit of 2 instead of 4. If both $\odot x_j^{\mathsf{T}}\odot$ and $\odot x_j^{\mathsf{F}}\odot$ exist for any token $X_j$, this implies that neither of $\odot x_{j'}^{\mathsf{T}}\odot$ and $\odot x_{j'}^{\mathsf{F}}\odot$ exists for some other $X_{j'}$, resulting in an uncompressed string. Then, we get at most a compression of $2fJ + 2f'(J-1) + 4I$:

$$\exists_{j\in\{1,\dots,J\}}\ |\mathcal{S}'\cap\{\odot x_j^{\mathsf{T}}\odot, \odot x_j^{\mathsf{F}}\odot\}| \neq 1 \implies \left(\texttt{toklen}(\mathcal{D},\mathcal{S}') \geq \underbrace{(4f + 3f')J + 2f' + I}_{\sum_{\mathbf{c}\in\mathcal{D}}|\mathbf{c}| - (2f+2f)'J + 2f' - 4I}\right) \tag{42}$$

By construction $f' = 2I + 1$, which leads to:

$$\exists_{j\in\{1,\dots,J\}}\ |\mathcal{S}'\cap\{\odot x_j^{\mathsf{T}}\odot, \odot x_j^{\mathsf{F}}\odot\}| \neq 1 \implies \left(\texttt{toklen}(\mathcal{D},\mathcal{S}') > (4f + 3f')J + 5I\right) \tag{43}$$

This concludes the proof. □

**LemmaProofStep 3.** (Step ③). *Any instance of the tokenisation problem with a solution which compresses the text by at least $(2f + 2f')J + 2\gamma$ symbols must be produced by a* `max-2-SAT` *problem with at least $\gamma$ satisfied clauses, i.e.,:*

$$\left(\texttt{toklen}(\mathcal{D},\mathcal{S}) \leq \underbrace{(4f + 3f')J + 5I - 2\gamma}_{\sum_{\mathbf{c}\in\mathcal{D}}|\mathbf{c}| - (2f+2f')J + 2\gamma}\right) \implies \text{M2S}(\mathcal{X},\mathcal{C},\gamma) \tag{44}$$

*Proof.* Finally, we now know any solution with this compression must have—for any variable $X_j$—either subword $\odot x_j^{\mathsf{T}}\odot$ or $\odot x_j^{\mathsf{F}}\odot$. We can thus create a bijection $\text{Conv}_{\mathcal{S}\to\chi}$ between the set of possible vocabularies respecting this condition, and the set of T/F assignments to SAT variables $\chi$:

$$\text{Conv}_{\mathcal{S}\to\chi}(\mathcal{S}) = \left\{ \begin{array}{ll} \texttt{T} & \text{if } \odot x_j^{\mathsf{T}}\odot \in \mathcal{S} \\ \texttt{F} & \text{if } \odot x_j^{\mathsf{F}}\odot \in \mathcal{S} \end{array} \right\}_{j=1}^{J} \tag{45}$$

Further, note that vocabularies of this form (as shown in Eq. (41)) lead to exactly $(2f + 2f')J$ symbols being compressed in $\mathcal{D}_1$ and $\mathcal{D}_2$. To achieve the target compression, a solution must thus compress $\mathcal{D}_3$ by at least $2\gamma$ symbols. Now note that for any string $\odot L_i^1 \odot L_i^2 \odot$ in $\mathcal{D}_3$ we have three compression options: $\odot L_i^1\odot$ will be compressed, saving 2 symbols; $\odot L_i^2\odot$ will be compressed, also saving 2 symbols; or nothing will be compressed. More specifically, $\odot L_i^1\odot$ can be compressed if $L_i^1$ represents $X_j$ and subword $\odot x_j^{\mathsf{T}}\odot$ exists, or if $L_i^1$ represents $\neg X_j$ and subword $\odot x_j^{\mathsf{F}}\odot$ exists; the same is true for $\odot L_i^2\odot$. They cannot both be compressed, however, as there is only one symbol $\odot$ between the literals. We thus get a compression of 2 symbols for each of these strings if at least one of its literals has an associated subword in $\mathcal{S}$. Note thus that whenever a string $\odot L_i^1 \odot L_i^2 \odot$ is compressed by 2 symbols using vocabulary $\mathcal{S}$, the `max-2-SAT` disjunction $L_i^1 \vee L_i^2$ will also be satisfied by assignment $\chi = \text{Conv}_{\mathcal{S}\to\chi}(\mathcal{S})$; similarly, whenever this string suffers no compression (i.e., having

a compression of zero), the max-2-SAT disjunction will not be satisfied. As our condition assumes a compression of at least $2\gamma$ symbols, we know that we have at least $\gamma$ strings for which a literal has an associated subword. We can thus write:

$$2\gamma \leq \max_{\mathcal{S} \subset \Sigma^+} \sum_{\mathbf{c} \in \mathcal{D}_3} |\mathbf{c}| - |\mathtt{tok}_\Downarrow[\mathcal{S}](\mathbf{c})| \tag{46a}$$

$$\text{s.t. } |\mathcal{S}| = |\Sigma| + J \text{ and } \forall_{j \in \{1,...,J\}} |\mathcal{S} \cap \{\odot x_j^\mathtt{T} \odot, \odot x_j^\mathtt{F} \odot\}| = 1$$

$$= \max_{\mathcal{S} \subset \Sigma^+} \sum_{\odot L_i^1 \odot L_i^2 \odot \, \in \mathcal{D}_3} 2 \, \mathbb{1} \left\{ \begin{array}{c} \odot L_i^1 \odot \in \mathcal{S} \\ \text{or} \\ \odot L_i^2 \odot \in \mathcal{S} \end{array} \right\} \tag{46b}$$

$$\text{s.t. } |\mathcal{S}| = |\Sigma| + J \text{ and } \forall_{j \in \{1,...,J\}} |\mathcal{S} \cap \{\odot x_j^\mathtt{T} \odot, \odot x_j^\mathtt{F} \odot\}| = 1$$

$$= \max_{\mathcal{X} \in \{0,1\}^J} \sum_{i=1}^I 2 \mathbb{1}_{\mathcal{X}} \{L_i^1 \vee L_i^2\} \tag{46c}$$

$$\implies \text{M2S}(\mathcal{X}, \mathcal{C}, \gamma) \tag{46d}$$

We thus know that, if a satisfying tokenisation solution exists, then the associated max-2-SAT problem will also be satisfiable. This concludes the proof. $\qquad\square$

## C. Proof of Lemma 6

**Lemma 6.** *If a* max-2-SAT *instance is satisfiable, then the bottom-up tokenisation instance output by Reduction 2 is also satisfiable. Formally:*

$$\text{M2S}(\mathcal{X}, \mathcal{C}, \gamma) \implies \text{Tok}_\uparrow(\text{R2}(\mathcal{X}, \mathcal{C}, \gamma)) \tag{23}$$

*Proof.* Our proof starts by first defining the three following lists of merges, which will be included in any satisfying solution to the tokenisation problem:

$$\mathbf{m}_1 = \bigcirc_{j=1}^J \left[ \langle \otimes, x_j^\mathtt{F} \rangle, \langle x_j^\mathtt{T}, \otimes \rangle \right], \quad \mathbf{m}_3 = \bigcirc_{j=1}^J \left[ \langle x_j^\mathtt{F}, \odot \rangle, \langle \odot, x_j^\mathtt{T} \rangle \right], \quad \mathbf{m}_5 = \bigcirc_{j=1}^J \left[ \langle \odot, x_j^\mathtt{F} \rangle, \langle x_j^\mathtt{T}, \odot \rangle \right] \tag{47}$$

Now, without loss of generality, let a satisfying solution to max-2-SAT have values $x_j^\star$. We then construct two other lists of merges, which depend on this max-2-SAT solution:

$$\mathbf{m}_2 = \bigcirc_{j=1}^J \left[ \begin{array}{cc} \langle \odot, x_j^\mathtt{T} \otimes \rangle & \text{if } x_j^\star = \mathtt{T} \\ \langle \otimes x_j^\mathtt{F}, \odot \rangle & \text{else} \end{array} \right], \quad \mathbf{m}_4 = \bigcirc_{j=1}^J \left[ \begin{array}{cc} \langle \odot x_j^\mathtt{T}, \odot \rangle & \text{if } x_j^\star = \mathtt{T} \\ \langle \odot, x_j^\mathtt{F} \odot \rangle & \text{else} \end{array} \right] \tag{48}$$

In words, we create merges $\langle \odot, x_j^\mathtt{T} \otimes \rangle$ and $\langle \odot x_j^\mathtt{T}, \odot \rangle$ if $x_j^\star$ is true, or $\langle \otimes x_j^\mathtt{F}, \odot \rangle$ and $\langle \odot, x_j^\mathtt{F} \odot \rangle$ if $x_j^\star$ is false. We then create a merge sequence by concatenating these lists in order:

$$\mathbf{m} = \mathbf{m}_1 \circ \mathbf{m}_2 \circ \mathbf{m}_3 \circ \mathbf{m}_4 \circ \mathbf{m}_5 \tag{49}$$

This gives us a total of $|\mathbf{m}| = K = 8J$ merges. Now we just need to count the symbols output by this solution to see if Eq. (17) is satisfied, since any given tokenisation $\mathtt{tok}_\uparrow[\mathbf{m}]$ will provide an upper bound on the optimal tokenisation in terms of compression:

$$\sum_{\mathbf{c} \in \mathcal{D}} |\mathtt{tok}_\uparrow[\mathbf{m}](\mathbf{c})| \geq \min_{\mathbf{m}' \in \mathcal{M}^*} \sum_{\mathbf{c} \in \mathcal{D}} |\mathtt{tok}_\uparrow[\mathbf{m}'](\mathbf{c})| \tag{50}$$

$$\text{s.t. } |\mathbf{m}'| = K$$

By applying the merges $\mathbf{m}$, each string in $\mathcal{D}_1$ will be compressed into a single subword; note that $\mathbf{m}_2$ and $\mathbf{m}_4$ will have no effect on these strings. This is easy to see by applying merges sequentially to these strings, as displayed in Tab. 1. leading to:

$$\sum_{\mathbf{c} \in (\bigcup_{=1}^f \mathcal{D}_1)} |\mathtt{tok}_\uparrow[\mathbf{m}](\mathbf{c})| = 6fJ \tag{51}$$

| $\mathbf{c}$ | $\mathrm{tok}_\uparrow[\mathbf{m}_1](\mathbf{c})$ | $\mathrm{tok}_\uparrow[\mathbf{m}_1 \circ \mathbf{m}_2 \circ \mathbf{m}_3](\mathbf{c})$ | $\mathrm{tok}_\uparrow[\mathbf{m}_1 \circ \mathbf{m}_2 \circ \mathbf{m}_3 \circ \mathbf{m}_4 \circ \mathbf{m}_5](\mathbf{c})$ | $|\mathrm{tok}_\uparrow[\mathbf{m}](\mathbf{c})|$ |
|---|---|---|---|---|
| $\langle \odot, x_j^{\mathtt{T}} \rangle$ | · | $\langle \odot x_j^{\mathtt{T}} \rangle$ | · | 1 |
| $\langle x_j^{\mathtt{F}}, \odot \rangle$ | · | $\langle x_j^{\mathtt{F}} \odot \rangle$ | · | 1 |
| $\langle x_j^{\mathtt{T}}, \odot \rangle$ | · | · | $\langle x_j^{\mathtt{T}} \odot \rangle$ | 1 |
| $\langle \odot, x_j^{\mathtt{F}} \rangle$ | · | · | $\langle \odot x_j^{\mathtt{F}} \rangle$ | 1 |
| $\langle x_j^{\mathtt{T}}, \otimes \rangle$ | $\langle x_j^{\mathtt{T}} \otimes \rangle$ | · | · | 1 |
| $\langle \otimes, x_j^{\mathtt{F}} \rangle$ | $\langle \otimes x_j^{\mathtt{F}} \rangle$ | · | · | 1 |

Table 1: Example of applying $\mathbf{m}$ in $\mathcal{D}_1$ of bottom-up tokenisation problem obtained from Reduction 2. The dot symbol · denotes the string not changing under the given merge.

| $\mathbf{c}$ | $\mathrm{tok}_\uparrow[\mathbf{m}_1](\mathbf{c})$ | $\mathrm{tok}_\uparrow[\mathbf{m}_1 \circ \mathbf{m}_2](\mathbf{c})$ | | $\mathrm{tok}_\uparrow[\mathbf{m}_1 \circ \mathbf{m}_2 \circ \mathbf{m}_3](\mathbf{c})$ | $\mathrm{tok}_\uparrow[\mathbf{m}_1 \circ \mathbf{m}_2 \circ \mathbf{m}_3 \circ \mathbf{m}_4](\mathbf{c})$ | | $|\mathrm{tok}_\uparrow[\mathbf{m}](\mathbf{c})|$ | |
|---|---|---|---|---|---|---|---|---|
| | | $x_j^\star = \mathtt{T}$ | $x_j^\star = \mathtt{F}$ | | $x_j^\star = \mathtt{T}$ | $x_j^\star = \mathtt{F}$ | $x_j^\star = \mathtt{T}$ | $x_j^\star = \mathtt{F}$ |
| $\langle \odot, x_j^{\mathtt{T}}, \odot \rangle$ | · | · | · | $\langle \odot x_j^{\mathtt{T}}, \odot \rangle$ | $\langle \odot x_j^{\mathtt{T}} \odot \rangle$ | $\langle \odot x_j^{\mathtt{T}}, \odot \rangle$ | 1 | 2 |
| $\langle \odot, x_j^{\mathtt{F}}, \odot \rangle$ | · | · | · | $\langle \odot, x_j^{\mathtt{F}} \odot \rangle$ | $\langle \odot, x_j^{\mathtt{F}} \odot \rangle$ | $\langle \odot x_j^{\mathtt{F}} \odot \rangle$ | 2 | 1 |
| $\langle \odot, x_j^{\mathtt{T}}, \otimes \rangle$ | $\langle \odot, x_j^{\mathtt{T}} \otimes \rangle$ | $\langle \odot x_j^{\mathtt{T}} \otimes \rangle$ | $\langle \odot, x_j^{\mathtt{T}} \otimes \rangle$ | · | · | · | 1 | 2 |
| $\langle \otimes, x_j^{\mathtt{F}}, \odot \rangle$ | $\langle \otimes x_j^{\mathtt{F}}, \odot \rangle$ | $\langle \otimes x_j^{\mathtt{F}}, \odot \rangle$ | $\langle \otimes x_j^{\mathtt{F}} \odot \rangle$ | · | · | · | 2 | 1 |

Table 2: Example of applying $\mathbf{m}$ in $\mathcal{D}_2$ of bottom-up tokenisation problem obtained from Reduction 2. The dot symbol · denotes the string not changing under the given merge. $\mathbf{m}_5$ is not depicted as it does not affect this dataset.

For each pair of strings $\odot x_j^{\mathtt{T}} \odot$ and $\odot x_j^{\mathtt{F}} \odot$ in $\mathcal{D}_2$, one is compressed into a single subword while the other is only compressed to two subwords—the one with $x_j^{\mathtt{T}}$ is compressed to a single symbol if $x_j^\star = \mathtt{T}$ and the one with $x_j^{\mathtt{F}}$ otherwise. The same is true for each pair of strings $\odot x_j^{\mathtt{T}} \otimes$ and $\otimes x_j^{\mathtt{F}} \odot$, also in $\mathcal{D}_2$. This is displayed in Tab. 2. We thus have that, for each variable $X_j$, the strings in $\mathcal{D}_2$ will occupy a total of $(1 + 2 + 1 + 2)J$ characters, and:

$$\sum_{\mathbf{c}\in(\bigcup_{\ell=1}^{f'}\mathcal{D}_2)} |\mathrm{tok}_\uparrow[\mathbf{m}](\mathbf{c})| = 6f'J \tag{52}$$

Similarly, each string in $\mathcal{D}_3$ and $\mathcal{D}_4$ will be compressed into only 2 symbols after this tokeniser is applied to it. We thus have:

$$\sum_{\mathbf{c}\in(\bigcup_{\ell=1}^{f''}\mathcal{D}_3)} |\mathrm{tok}_\uparrow[\mathbf{m}](\mathbf{c})| = 4f''J, \qquad \sum_{\mathbf{c}\in(\bigcup_{\ell=1}^{f'''}\mathcal{D}_4)} |\mathrm{tok}_\uparrow[\mathbf{m}](\mathbf{c})| = 4f'''J \tag{53}$$

Finally, we have the strings in $\mathcal{D}_5$. These strings are constructed such that they will be compressed into 2 symbols if either $L_i^1$ or $L_i^2$ evaluates to $\mathtt{T}$, and kept with 3 symbols otherwise; see Tab. 4 for a detailed simulation of why this is the case. We thus have:

$$\sum_{\mathbf{c}\in\mathcal{D}_5} |\mathrm{tok}_\uparrow[\mathbf{m}](\mathbf{c})| = \sum_{i=1}^{I}\left(3 - 1\,\mathbb{1}\left\{\begin{array}{l} L_i^1 = X_j \quad \text{and } \langle \odot, x_j^{\mathtt{T}} \otimes \rangle, \langle \odot x_j^{\mathtt{T}}, \odot \rangle \in \mathbf{m} \\ \text{or} \\ L_i^1 = \neg X_j \ \text{and } \langle \otimes x_j^{\mathtt{F}}, \odot \rangle, \langle \odot, x_j^{\mathtt{F}} \odot \rangle \in \mathbf{m} \\ \text{or} \\ L_i^2 = X_{j'} \quad \text{and } \langle \odot, x_{j'}^{\mathtt{T}} \otimes \rangle, \langle \odot x_{j'}^{\mathtt{T}}, \odot \rangle \in \mathbf{m} \\ \text{or} \\ L_i^2 = \neg X_{j'} \ \text{and } \langle \otimes x_{j'}^{\mathtt{F}}, \odot \rangle, \langle \odot, x_{j'}^{\mathtt{F}} \odot \rangle \in \mathbf{m} \end{array}\right\}\right) \tag{54a}$$

$$= 3I - \sum_{i=1}^{I} \mathbb{1}_{\chi^\star}\{L_i^1 \vee L_i^2\} \tag{54b}$$

$$\leq 3I - \gamma \tag{54c}$$

where, by construction, we have a merge in our sequence (e.g., $\langle \odot, x_j^{\mathtt{T}} \otimes \rangle$ or $\langle \otimes x_j^{\mathtt{F}}, \odot \rangle$) if and only if its value is in a satisfying assignment (e.g., $x_j^\star = \mathtt{T}$ or $x_j^\star = \mathtt{F}$ respectively). Summing together the lengths in Eqs. (51) to (54), we get that:

$$\sum_{\mathbf{c}\in\mathcal{D}} |\mathrm{tok}_\uparrow[\mathbf{m}](\mathbf{c})| \leq \delta = \left(6f + 6f' + 4f'' + 4f'''\right)J + 3I - \gamma \tag{55}$$

| $\mathcal{D}$ | $\mathbf{c}$ | $\mathrm{tok}_\uparrow[\mathbf{m}_1](\mathbf{c})$ | $\mathrm{tok}_\uparrow[\mathbf{m}_1\circ\mathbf{m}_2](\mathbf{c})$ | | $\mathrm{tok}_\uparrow[\mathbf{m}_1\circ\mathbf{m}_2\circ\mathbf{m}_3](\mathbf{c})$ | | $\mathrm{tok}_\uparrow[\mathbf{m}_1\circ\mathbf{m}_2\circ\mathbf{m}_3\circ\mathbf{m}_4](\mathbf{c})$ | | $\mathrm{tok}_\uparrow[\mathbf{m}_1\circ\mathbf{m}_2\circ\mathbf{m}_3\circ\mathbf{m}_4\circ\mathbf{m}_5](\mathbf{c})$ | | $|\mathrm{tok}_\uparrow[\mathbf{m}](\mathbf{c})|$ |
|---|---|---|---|---|---|---|---|---|---|---|---|
| | | | $x_j^\star=\mathrm{T}$ | $x_j^\star=\mathrm{F}$ | $x_j^\star=\mathrm{T}$ | $x_j^\star=\mathrm{F}$ | $x_j^\star=\mathrm{T}$ | $x_j^\star=\mathrm{F}$ | $x_j^\star=\mathrm{T}$ | $x_j^\star=\mathrm{F}$ | |
| $\mathcal{D}_3$ | $\langle\odot,x_j^\mathrm{T},\odot,x_j^\mathrm{F},\odot\rangle$ | $\cdot$ | $\cdot$ | $\cdot$ | $\langle\odot x_j^\mathrm{T},\odot,x_j^\mathrm{F}\odot\rangle$ | | $\langle\odot x_j^\mathrm{T},\odot,x_j^\mathrm{F}\odot\rangle$ | $\langle\odot x_j^\mathrm{T},\odot x_j^\mathrm{F}\odot\rangle$ | $\cdot$ | $\cdot$ | 2 |
| $\mathcal{D}_3$ | $\langle\otimes,x_j^\mathrm{F},\odot,x_j^\mathrm{T},\otimes\rangle$ | $\langle\otimes x_j^\mathrm{F},\odot,x_j^\mathrm{T}\otimes\rangle$ | $\langle\otimes x_j^\mathrm{F},\odot x_j^\mathrm{T}\otimes\rangle$ | $\langle\otimes x_j^\mathrm{F}\odot,x_j^\mathrm{T}\otimes\rangle$ | $\cdot$ | | $\cdot$ | $\langle\odot x_j^\mathrm{F},\odot x_j^\mathrm{T}\otimes\rangle$ | $\langle\odot x_j^\mathrm{F},\odot x_j^\mathrm{T}\otimes\rangle$ | $\cdot$ | 2 |
| $\mathcal{D}_4$ | $\langle\odot,x_j^\mathrm{F},\odot,x_j^\mathrm{T},\otimes\rangle$ | $\langle\odot,x_j^\mathrm{F},\odot,x_j^\mathrm{T}\otimes\rangle$ | $\langle\odot,x_j^\mathrm{F},\odot x_j^\mathrm{T}\otimes\rangle$ | $\cdot$ | $\cdot$ | $\langle\odot,x_j^\mathrm{F},\odot x_j^\mathrm{T}\otimes\rangle$ | $\cdot$ | $\langle\otimes x_j^\mathrm{F}\odot,x_j^\mathrm{T}\otimes\rangle$ | $\langle\otimes x_j^\mathrm{F},\odot x_j^\mathrm{T}\otimes\rangle$ | $\cdot$ | 2 |
| $\mathcal{D}_4$ | $\langle\otimes,x_j^\mathrm{F},\odot,x_j^\mathrm{T},\odot\rangle$ | $\langle\otimes x_j^\mathrm{F},\odot,x_j^\mathrm{T},\odot\rangle$ | $\cdot$ | $\langle\otimes x_j^\mathrm{F}\odot,x_j^\mathrm{T},\odot\rangle$ | $\langle\otimes x_j^\mathrm{F},\odot x_j^\mathrm{T},\odot\rangle$ | | $\langle\otimes x_j^\mathrm{F},\odot x_j^\mathrm{T}\odot\rangle$ | $\cdot$ | $\cdot$ | $\langle\otimes x_j^\mathrm{F}\odot,x_j^\mathrm{T}\odot\rangle$ | 2 |

Table 3: Example of applying $\mathbf{m}$ in $\mathcal{D}_3$ and $\mathcal{D}_4$ of the bottom-up tokenisation problem obtained from Reduction 2. The dot symbol $\cdot$ denotes the string not changing under the given merge.

| Assignment | Condition | $\mathbf{c}$ | $\mathrm{tok}_\uparrow[\mathbf{m}_1](\mathbf{c})$ | $\mathrm{tok}_\uparrow[\mathbf{m}_1\circ\mathbf{m}_2](\mathbf{c})$ | $\mathrm{tok}_\uparrow[\mathbf{m}_1\circ\mathbf{m}_2\circ\mathbf{m}_3](\mathbf{c})$ | $\mathrm{tok}_\uparrow[\mathbf{m}_1\circ\mathbf{m}_2\circ\mathbf{m}_3\circ\mathbf{m}_4](\mathbf{c})$ | $|\mathrm{tok}_\uparrow[\mathbf{m}](\mathbf{c})|$ |
|---|---|---|---|---|---|---|---|
| $L_i^1=X_j$ and $L_i^2=\neg X_{j'}$ | $x_j^\star=\mathrm{T}\wedge x_{j'}^\star=\mathrm{T}$ $x_j^\star=\mathrm{F}\wedge x_{j'}^\star=\mathrm{T}$ $x_j^\star=\mathrm{T}\wedge x_{j'}^\star=\mathrm{F}$ $x_j^\star=\mathrm{F}\wedge x_{j'}^\star=\mathrm{F}$ | $\langle\odot,x_j^\mathrm{T},\odot,x_{j'}^\mathrm{F},\odot\rangle$ | $\cdot$ | $\cdot$ | $\langle\odot x_j^\mathrm{T},\odot,x_{j'}^\mathrm{F}\odot\rangle$ | $\langle\odot x_j^\mathrm{T}\odot,x_{j'}^\mathrm{F},\odot\rangle$ $\langle\odot x_j^\mathrm{T}\odot,x_{j'}^\mathrm{F},\odot\rangle$ $\langle\odot x_j^\mathrm{T}\odot,x_{j'}^\mathrm{F},\odot\rangle$ $\langle\odot x_j^\mathrm{T},\odot x_{j'}^\mathrm{F}\odot\rangle$ | 2 3 2 2 |
| $L_i^1=\neg X_j$ and $L_i^2=X_{j'}$ | $x_j^\star=\mathrm{T}\wedge x_{j'}^\star=\mathrm{T}$ $x_j^\star=\mathrm{F}\wedge x_{j'}^\star=\mathrm{T}$ $x_j^\star=\mathrm{T}\wedge x_{j'}^\star=\mathrm{F}$ $x_j^\star=\mathrm{F}\wedge x_{j'}^\star=\mathrm{F}$ | $\langle\odot,x_j^\mathrm{T},\odot,x_{j'}^\mathrm{F},\odot\rangle$ | $\cdot$ | $\cdot$ | $\langle\odot x_j^\mathrm{T},\odot,x_{j'}^\mathrm{F}\odot\rangle$ | $\langle\odot x_{j'}^\mathrm{T}\odot,x_j^\mathrm{F}\odot\rangle$ $\langle\odot x_{j'}^\mathrm{T}\odot,x_j^\mathrm{F}\odot\rangle$ $\langle\odot x_{j'}^\mathrm{T},\odot,x_j^\mathrm{F}\odot\rangle$ $\langle\odot x_{j'}^\mathrm{T},\odot x_j^\mathrm{F}\odot\rangle$ | 2 2 3 2 |
| $L_i^1=\neg X_j$ and $L_i^2=\neg X_{j'}$ | $x_j^\star=\mathrm{T}\wedge x_{j'}^\star=\mathrm{T}$ $x_j^\star=\mathrm{F}\wedge x_{j'}^\star=\mathrm{T}$ $x_j^\star=\mathrm{T}\wedge x_{j'}^\star=\mathrm{F}$ $x_j^\star=\mathrm{F}\wedge x_{j'}^\star=\mathrm{F}$ | $\langle\otimes,x_j^\mathrm{F},\odot,x_{j'}^\mathrm{F},\odot\rangle$ | $\langle\otimes x_j^\mathrm{F},\odot,x_{j'}^\mathrm{F},\odot\rangle$ | $\langle\otimes x_j^\mathrm{F}\odot,x_{j'}^\mathrm{F},\odot\rangle$ $\langle\otimes x_j^\mathrm{F}\odot,x_{j'}^\mathrm{F},\odot\rangle$ | $\langle\otimes x_j^\mathrm{F},\odot,x_{j'}^\mathrm{F}\odot\rangle$ $\langle\otimes x_j^\mathrm{F}\odot,x_{j'}^\mathrm{F}\odot\rangle$ $\langle\otimes x_j^\mathrm{F},\odot,x_{j'}^\mathrm{F}\odot\rangle$ $\langle\otimes x_j^\mathrm{F}\odot,x_{j'}^\mathrm{F}\odot\rangle$ | $\langle\otimes x_j^\mathrm{F},\odot x_{j'}^\mathrm{F},\odot\rangle$ | 3 2 2 2 |
| $L_i^1=X_j$ and $L_i^2=X_{j'}$ | $x_j^\star=\mathrm{T}\wedge x_{j'}^\star=\mathrm{T}$ $x_j^\star=\mathrm{F}\wedge x_{j'}^\star=\mathrm{T}$ $x_j^\star=\mathrm{T}\wedge x_{j'}^\star=\mathrm{F}$ $x_j^\star=\mathrm{F}\wedge x_{j'}^\star=\mathrm{F}$ | $\langle\odot,x_j^\mathrm{T},\odot,x_{j'}^\mathrm{T},\otimes\rangle$ | $\langle\odot,x_j^\mathrm{T},\odot,x_{j'}^\mathrm{T},\otimes\rangle$ | $\langle\odot,x_j^\mathrm{T},\odot x_{j'}^\mathrm{T}\otimes\rangle$ $\langle\odot x_j^\mathrm{T},\odot x_{j'}^\mathrm{T}\otimes\rangle$ $\cdot$ | $\langle\odot x_j^\mathrm{T},\odot x_{j'}^\mathrm{T}\otimes\rangle$ $\langle\odot x_j^\mathrm{T},\odot x_{j'}^\mathrm{T}\otimes\rangle$ $\langle\odot x_j^\mathrm{T}\odot,x_{j'}^\mathrm{T}\otimes\rangle$ | 2 2 2 3 |

Table 4: Example of applying $\mathbf{m}$ in $\mathcal{D}_5$ of the bottom-up tokenisation problem obtained from Reduction 2. The dot symbol $\cdot$ denotes the string not changing under the given merge. $\mathbf{m}_5$ is not depicted as it does not affect this dataset.

which concludes the proof.

$\square$

# D. Proof of Lemma 7

**Lemma 7.** *If the bottom-up tokenisation instance output by Reduction 2 is satisfiable, the* max-2-SAT *instance which generated it is as well. Formally:*

$$\mathrm{Tok}_\uparrow(\mathrm{R2}(\mathcal{X},\mathcal{C},\gamma)) \implies \mathrm{M2S}(\mathcal{X},\mathcal{C},\gamma) \tag{27}$$

*Proof.* First, note that:

$$\sum_{\mathbf{c}\in\mathcal{D}}|\mathbf{c}| = (12f+12f'+10f''+10f''')\,J + 5\,I \tag{56}$$

Further, let:

$$\texttt{toklen}(\mathcal{D},\mathbf{m}) \overset{\text{def}}{=} \sum_{\mathbf{c}\in\mathcal{D}}|\mathrm{tok}_\uparrow[\mathbf{m}](\mathbf{c})| \tag{57}$$

$$\mathbf{m}_1 = \bigcirc_{j=1}^{J}[\langle\otimes,x_j^\mathrm{F}\rangle,\langle x_j^\mathrm{T},\otimes\rangle], \qquad \mathbf{m}_3 = \bigcirc_{j=1}^{J}[\langle x_j^\mathrm{F},\odot\rangle,\langle\odot,x_j^\mathrm{T}\rangle], \qquad \mathbf{m}_5 = \bigcirc_{j=1}^{J}[\langle\odot,x_j^\mathrm{F}\rangle,\langle x_j^\mathrm{T},\odot\rangle]$$

$$\mathbf{m}_j^\odot = \left\{\begin{array}{l}\langle\odot x_j^\mathrm{T},\odot\rangle,\langle\odot,x_j^\mathrm{F}\odot\rangle\\\langle\odot,x_j^\mathrm{T}\odot\rangle,\langle\odot x_j^\mathrm{F},\odot\rangle\end{array}\right\}, \qquad \mathbf{m}_j^\otimes = \left\{\begin{array}{l}\langle\odot,x_j^\mathrm{T}\otimes\rangle,\langle\otimes x_j^\mathrm{F},\odot\rangle\\\langle\odot x_j^\mathrm{T},\otimes\rangle,\langle\otimes,x_j^\mathrm{F}\odot\rangle\end{array}\right\}$$

$$\mathbf{m}_j^\mathrm{T} = \left\{\begin{array}{l}\langle\odot x_j^\mathrm{T},\odot\rangle,\langle\odot,x_j^\mathrm{T}\otimes\rangle,\\\langle\odot,x_j^\mathrm{T}\odot\rangle,\langle\odot x_j^\mathrm{T},\otimes\rangle,\end{array}\right\}, \qquad \mathbf{m}_j^\mathrm{F} = \left\{\begin{array}{l}\langle\odot,x_j^\mathrm{F}\odot\rangle,\langle\otimes x_j^\mathrm{F},\odot\rangle,\\\langle\odot x_j^\mathrm{F},\odot\rangle,\langle\otimes,x_j^\mathrm{F}\odot\rangle,\end{array}\right\}$$

The maximum number of symbols in this dataset after compression is set to $\delta = (6f+6f'+4f''+4f''')\,J+3\,I-\gamma$. This means that to satisfy this objective, there must exist a vocabulary whose tokeniser compresses the text by at least $(6f+6f'+6f''+6f''')\,J+2I+\gamma$ symbols. We now prove this lemma in five steps: ① we show that any solution which compresses the text by at least $6fJ$ symbols must include all merges in $\mathbf{m}_1$, $\mathbf{m}_3$, and $\mathbf{m}_5$; ② we show that any solution

which compresses the text by at least $(6f + 6f')J$ symbols must only include either merges in $\mathbf{m}_1, \mathbf{m}_3, \mathbf{m}_5$, or in either $\mathbf{m}_j^{\odot}$ or $\mathbf{m}_j^{\otimes}$; ③ we show that any solution which compresses the text by at least $(6f + 6f' + 6f'')J$ symbols must include, for each $j \in \{1, \ldots, J\}$, exactly one merge in set $\mathbf{m}_j^{\odot}$ and one in set $\mathbf{m}_j^{\otimes}$; ④ we show that any solution which compresses the text by at least $(6f + 6f' + 6f'' + 6f''')J$ symbols must include, for each $j \in \{1, \ldots, J\}$, exactly two merges in either set $\mathbf{m}_j^{\mathsf{T}}$ or in set $\mathbf{m}_j^{\mathsf{F}}$; ⑤ we show that any solution which compresses the text by at least $(6f + 6f' + 6f'' + 6f''')J + 2I + \gamma$ symbols must be produced by a `max-2-SAT` problem with at least $\gamma$ satisfied clauses. $\square$

**LemmaProofStep 1.** (Step ①). *Any solution which compresses the text by at least $6fJ$ symbols must include all merges in* $\mathbf{m}_1, \mathbf{m}_3,$ *and* $\mathbf{m}_5$, *i.e.,:*

$$\left( \texttt{toklen}(\mathcal{D}, \mathbf{m}) \leq \underbrace{6fJ + \left(12f' + 10f'' + 10f'''\right)J + 5I}_{\sum_{\mathbf{c} \in \mathcal{D}} |\mathbf{c}| - 6fJ} \right) \tag{58}$$

$$\implies \underbrace{\bigcirc_{j=1}^{J}[\langle \otimes, x_j^{\mathsf{F}} \rangle, \langle x_j^{\mathsf{T}}, \otimes \rangle]}_{\mathbf{m}_1} \subset \mathbf{m}, \quad \underbrace{\bigcirc_{j=1}^{J}[\langle x_j^{\mathsf{F}}, \odot \rangle, \langle \odot, x_j^{\mathsf{T}} \rangle]}_{\mathbf{m}_3} \subset \mathbf{m}, \quad \underbrace{\bigcirc_{j=1}^{J}[\langle \odot, x_j^{\mathsf{F}} \rangle, \langle x_j^{\mathsf{T}}, \odot \rangle]}_{\mathbf{m}_5} \subset \mathbf{m}$$

*Proof.* We prove this statement by contradiction. Assume that one of the subwords above is not present in the tokenisers' merge sequence $\mathbf{m}$. In that case, the strings in $\mathcal{D}_1$ which contain this character string will not be compressed, and will thus still be represented with 2 symbols. There will thus be at most $6J - 1$ strings in $\mathcal{D}_1$ represented with a single symbol, and at least one represented with two symbols. The minimum length achievable would thus be:

$$\texttt{toklen}(\mathcal{D}, \mathbf{m}) = \underbrace{\sum_{\mathbf{c} \in \bigcup_{-=1}^{f} \mathcal{D}_1} |\texttt{tok}_\uparrow[\mathbf{m}](\mathbf{c})|}_{\geq (6J-1)f + 2f} + \underbrace{\sum_{\mathbf{c} \in \mathcal{D} \setminus (\bigcup_{-=1}^{f} \mathcal{D}_1)} |\texttt{tok}_\uparrow[\mathbf{m}](\mathbf{c})|}_{>0} \tag{59a}$$

$$> (6J + 1)f \qquad \text{By construction } f = \left(12f' + 10f'' + 10f'''\right)J + 5I \tag{59b}$$

$$= \left(6f + 12f' + 10f'' + 10f'''\right)J + 5I \tag{59c}$$

which contradicts the proof's statement. $\square$

**LemmaProofStep 2.** (Step ②). *Any solution which compresses the text by at least $(6f + 6f')J$ symbols must only include either merges in* $\mathbf{m}_1, \mathbf{m}_3, \mathbf{m}_5$, *or in either* $\mathbf{m}_j^{\odot}$ *or* $\mathbf{m}_j^{\otimes}$, *i.e.,:*

$$\left( \texttt{toklen}(\mathcal{D}, \mathcal{S}) \leq \underbrace{(6f + 6f')J + \left(10f'' + 10f'''\right)J + 5I}_{\sum_{\mathbf{c} \in \mathcal{D}} |\mathbf{c}| - (6f + 6f')J} \right) \tag{60}$$

$$\implies \mathbf{m} \setminus (\mathbf{m}_1 \circ \mathbf{m}_3 \circ \mathbf{m}_5) \subseteq \underbrace{\left\{ \begin{array}{l} \langle \odot, x_j^{\mathsf{T}} \otimes \rangle, \langle \otimes x_j^{\mathsf{F}}, \odot \rangle, \langle \odot x_j^{\mathsf{T}}, \odot \rangle, \langle \odot, x_j^{\mathsf{F}} \odot \rangle \\ \langle \odot x_j^{\mathsf{T}}, \otimes \rangle, \langle \otimes, x_j^{\mathsf{F}} \odot \rangle, \langle \odot, x_j^{\mathsf{F}} \odot \rangle, \langle \odot x_j^{\mathsf{F}}, \odot \rangle \end{array} \right\}_{j=1}^{J}}_{\bigcup_{j=1}^{J}(\mathbf{m}_j^{\odot} \cup \mathbf{m}_j^{\otimes})}$$

*Proof.* We again prove this statement by contradiction. Assume that $\mathbf{m}$ has all merges $\mathbf{m}_1, \mathbf{m}_3, \mathbf{m}_5$, but one of its other merges is in neither of the sets $\mathbf{m}_j^{\odot}$ and $\mathbf{m}_j^{\otimes}$. This means that at least one of the sets $\mathbf{m}_j^{\odot} \cup \mathbf{m}_j^{\otimes}$ will not have at least two merges in the solution; this is because there are $J$ such sets ($\mathbf{m}_j^{\odot} \cup \mathbf{m}_j^{\otimes}$), which—coupled together with the $6J$ already selected merges in $\mathbf{m}_1, \mathbf{m}_3, \mathbf{m}_5$—would amount to the maximum of $8J$ merges. In that case, the strings (e.g., $\odot x_j^{\mathsf{T}} \odot$, $\odot x_j^{\mathsf{F}} \odot$, $\odot x_j^{\mathsf{T}} \otimes$ and $\otimes x_j^{\mathsf{F}} \odot$) in $\mathcal{D}_2$ containing the characters this absent merge represents will not be fully compressed to a single symbol, being represented with 2 symbols instead. Out of the $4J$ strings in $\mathcal{D}_2$ then, there will thus be at most $2J - 1$ represented with a single symbol, and at least $2J + 1$ represented with two symbols—resulting in a total of at least $2J - 1 + 2(2J + 1) = 6J + 1$ symbols. The minimum length achievable would thus be:

$$\texttt{toklen}(\mathcal{D}, \mathbf{m}) = \underbrace{\sum_{\mathbf{c} \in \bigcup_{-=1}^{f} \mathcal{D}_1} |\texttt{tok}_\uparrow[\mathbf{m}](\mathbf{c})|}_{= 6fJ} + \underbrace{\sum_{\mathbf{c} \in \bigcup_{-=1}^{f'} \mathcal{D}_2} |\texttt{tok}_\uparrow[\mathbf{m}](\mathbf{c})|}_{\geq (6J+1)f'} + \underbrace{\sum_{\mathbf{c} \in \mathcal{D} \setminus (\mathcal{D}_1 \cup \mathcal{D}_2)} |\texttt{tok}_\uparrow[\mathbf{m}](\mathbf{c})|}_{>0} \tag{61a}$$

$$> 6fJ + (6J + 1)f' \qquad \text{By construction } f' = \left(10f'' + 10f'''\right)J + 5I \tag{61b}$$

$$= \left(6f + 6f' + 10f'' + 10f'''\right)J + 5I \tag{61c}$$

which contradicts the proof's statement. □

**LemmaProofStep 3.** (Step ③). *Any solution which compresses the text by at least $(6f + 6f' + 6f'')J$ symbols must include all merges in $\mathbf{m}_1$, $\mathbf{m}_3$, $\mathbf{m}_5$, and, for each $j \in \{1, \ldots, J\}$, exactly one merge in set $\mathbf{m}_j^{\circledcirc}$ and one in set $\mathbf{m}_j^{\otimes}$, i.e.,:*

$$\left( \texttt{toklen}(\mathcal{D}, \mathbf{m}) \leq \underbrace{(6f + 6f' + 4f'')J + 10f'''J + 5I}_{\sum_{\mathbf{c} \in \mathcal{D}} |\mathbf{c}| - (6f + 6f' + 6f'')J} \right) \tag{62}$$

$$\implies \forall_{j \in \{1, \ldots, J\}} \left| \mathbf{m} \cap \underbrace{\left\{ \begin{matrix} \langle \odot x_j^{\mathsf{T}}, \odot \rangle, \langle \odot, x_j^{\mathsf{F}} \odot \rangle \\ \langle \odot, x_j^{\mathsf{T}} \odot \rangle, \langle \odot x_j^{\mathsf{F}}, \odot \rangle \end{matrix} \right\}}_{\mathbf{m}_j^{\circledcirc}} \right| = 1 \text{ and } \left| \mathbf{m} \cap \underbrace{\left\{ \begin{matrix} \langle \odot, x_j^{\mathsf{T}} \otimes \rangle, \langle \otimes x_j^{\mathsf{F}}, \odot \rangle \\ \langle \odot x_j^{\mathsf{T}}, \otimes \rangle, \langle \otimes, x_j^{\mathsf{F}} \odot \rangle \end{matrix} \right\}}_{\mathbf{m}_j^{\otimes}} \right| = 1$$

*Proof.* We again prove this statement by contradiction. First, assume that $\mathbf{m}$ contains all the merges in $\mathbf{m}_1, \mathbf{m}_3, \mathbf{m}_5$; further, assume all its other merges are contained in sets $\mathbf{m}_j^{\circledcirc}$ and $\mathbf{m}_j^{\otimes}$. Note now that, if any merge in $\mathbf{m}_j^{\otimes}$ is in the selected merges $\mathbf{m}$, the string $\otimes x_j^{\mathsf{F}} \odot x_j^{\mathsf{T}} \otimes$ in $\mathcal{D}_3$ will be compressed to 2 symbols (e.g., $\langle \otimes x_j^{\mathsf{F}}, \odot x_j^{\mathsf{T}} \otimes \rangle$); if none of these merges is present, however, this string will only be compressed to 3 symbols (e.g., $\langle \otimes x_j^{\mathsf{F}}, \odot, x_j^{\mathsf{T}} \otimes \rangle$). The same is true for strings $\odot x_j^{\mathsf{T}} \odot x_j^{\mathsf{F}} \odot$ and merges in $\mathbf{m}_j^{\circledcirc}$. Now, assume the contradictory case: for a value of $j \in \{1, \ldots, J\}$, $\mathbf{m}$ does not satisfy the condition above. As, by construction, our solution has $K = 8J$ merges, and because $|\mathbf{m}_1 \circ \mathbf{m}_3 \circ \mathbf{m}_5| = 6J$, we know that we have $2J$ merges in sets $\mathbf{m}_j^{\circledcirc}$ and $\mathbf{m}_j^{\otimes}$. As there are exactly $2J$ such sets, if the condition above does not hold, at least one of these sets must have no merge present in $\mathbf{m}$. In that case, the strings in $\mathcal{D}_3$ which contain the character string represented by these absent merges will be compressed to three symbols, while others will be compressed to two symbols. There will thus be at most $2J - 1$ strings in $\mathcal{D}_3$ represented with two symbols, and at least one represented with three symbols. The minimum length achievable would thus be:

$$\texttt{toklen}(\mathcal{D}, \mathbf{m}) = \underbrace{\sum_{\mathbf{c} \in \bigcup_{\underline{\cdot}=1}^{f} \mathcal{D}_1 \cup \bigcup_{\underline{\cdot}=1}^{f'} \mathcal{D}_2} |\texttt{tok}_\uparrow[\mathbf{m}](\mathbf{c})|}_{=(6f + 6f')J} + \underbrace{\sum_{\mathbf{c} \in \bigcup_{\underline{\cdot}=1}^{f''} \mathcal{D}_3} |\texttt{tok}_\uparrow[\mathbf{m}](\mathbf{c})|}_{\geq (2J-1)2f'' + 3f''} + \underbrace{\sum_{\mathbf{c} \in \bigcup_{\underline{\cdot}=1}^{f'''} \mathcal{D}_4 \cup \mathcal{D}_5} |\texttt{tok}_\uparrow[\mathbf{m}](\mathbf{c})|}_{>0} \tag{63a}$$

$$> (6f + 6f')J + (4J + 1)f'' \qquad \text{By construction } f'' = 10f'''J + 5I \tag{63b}$$

$$= (6f + 6f' + 4f'' + 10f''')J + 5I \tag{63c}$$

which contradicts the proof's statement. □

**LemmaProofStep 4.** (Step ④). *Any solution which compresses the text by at least $(6f + 6f' + 6f'' + 6f''')J$ symbols must include all merges in $\mathbf{m}_1$, $\mathbf{m}_3$, $\mathbf{m}_5$, and, for each $j \in \{1, \ldots, J\}$, exactly one merge in set $\mathbf{m}_j^{\circledcirc}$ and one in set $\mathbf{m}_j^{\otimes}$, such that either both these merges are in $\mathbf{m}_j^{\mathsf{T}}$ or both are in $\mathbf{m}_j^{\mathsf{F}}$, i.e.,:*

$$\left( \texttt{toklen}(\mathcal{D}, \mathbf{m}) \leq \underbrace{(6f + 6f' + 4f'' + 4f''')J + 5I}_{\sum_{\mathbf{c} \in \mathcal{D}} |\mathbf{c}| - (6f + 6f' + 6f'' + 6f''')J} \right) \tag{64}$$

$$\implies \forall_{j \in \{1, \ldots, J\}} \left| \mathbf{m} \cap \underbrace{\left\{ \begin{matrix} \langle \odot x_j^{\mathsf{T}}, \odot \rangle, \langle \odot, x_j^{\mathsf{T}} \otimes \rangle, \\ \langle \odot, x_j^{\mathsf{T}} \odot \rangle, \langle \odot x_j^{\mathsf{T}}, \otimes \rangle, \end{matrix} \right\}}_{\mathbf{m}_j^{\mathsf{T}}} \right| = 2 \text{ or } \left| \mathbf{m} \cap \underbrace{\left\{ \begin{matrix} \langle \odot, x_j^{\mathsf{F}} \odot \rangle, \langle \otimes x_j^{\mathsf{F}}, \odot \rangle, \\ \langle \odot x_j^{\mathsf{F}}, \odot \rangle, \langle \otimes, x_j^{\mathsf{F}} \odot \rangle, \end{matrix} \right\}}_{\mathbf{m}_j^{\mathsf{F}}} \right| = 2$$

*Proof.* First, note that the conditions of the step of our proof are stricter than previous ones, so we assume the conditions of steps ① to ③ hold—i.e., $\mathbf{m}$ contains all merges in $\mathbf{m}_1, \mathbf{m}_3, \mathbf{m}_5$; further, it has one and only one merge from each set $\mathbf{m}_j^{\circledcirc}$ and $\mathbf{m}_j^{\otimes}$. (Note that $\mathbf{m}_j^{\circledcirc} \cup \mathbf{m}_j^{\otimes} = \mathbf{m}_j^{\mathsf{T}} \cup \mathbf{m}_j^{\mathsf{F}}$, and that the just-mentioned condition implies $|\mathbf{m} \cap (\mathbf{m}_j^{\mathsf{T}} \cup \mathbf{m}_j^{\mathsf{F}})| = 2$.) We now again prove this statement by contradiction. Consider now the case:

$$\left| \mathbf{m} \cap \underbrace{\left\{ \begin{matrix} \langle \odot x_j^{\mathsf{T}}, \odot \rangle, \langle \odot, x_j^{\mathsf{T}} \otimes \rangle, \\ \langle \odot, x_j^{\mathsf{T}} \odot \rangle, \langle \odot x_j^{\mathsf{T}}, \otimes \rangle, \end{matrix} \right\}}_{\mathbf{m}_j^{\mathsf{T}}} \right| = 2 \text{ or } \left| \mathbf{m} \cap \underbrace{\left\{ \begin{matrix} \langle \odot, x_j^{\mathsf{F}} \odot \rangle, \langle \otimes x_j^{\mathsf{F}}, \odot \rangle, \\ \langle \odot x_j^{\mathsf{F}}, \odot \rangle, \langle \otimes, x_j^{\mathsf{F}} \odot \rangle, \end{matrix} \right\}}_{\mathbf{m}_j^{\mathsf{F}}} \right| = 2 \tag{65}$$

If this is true, then strings $\odot x_j^{\mathrm{F}} \odot x_j^{\mathrm{T}} \otimes$ and $\otimes x_j^{\mathrm{F}} \odot x_j^{\mathrm{T}} \odot$ in $\mathcal{D}_4$ will be compressed to 2 symbols each (e.g., to $\langle \odot x_j^{\mathrm{F}}, \odot x_j^{\mathrm{T}} \otimes \rangle$ and $\langle \otimes x_j^{\mathrm{F}}, \odot x_j^{\mathrm{T}} \odot \rangle$ or $\langle \odot x_j^{\mathrm{F}} \odot, x_j^{\mathrm{T}} \otimes \rangle$ and $\langle \otimes x_j^{\mathrm{F}} \odot, x_j^{\mathrm{T}} \odot \rangle$ ); if this condition is false, however, one of these strings will only be compressed to 3 symbols (e.g., to $\langle \odot x_j^{\mathrm{F}}, \odot x_j^{\mathrm{T}} \otimes \rangle$ and $\langle \otimes x_j^{\mathrm{F}}, \odot, x_j^{\mathrm{T}} \odot \rangle$). Now, assume the contradictory case: for a value of $j \in \{1, \ldots, J\}$, $\mathbf{m}$ does not satisfy the condition above. In that case, the strings in $\mathcal{D}_4$ for which the condition does not hold will be compressed to $3 + 2$ symbols, while others will be compressed to $2 + 2$ symbols. There will thus be at most $2J - 1$ strings in $\mathcal{D}_4$ represented with two symbols, and at least one represented with three symbols. The minimum length achievable would thus be:

$$\texttt{toklen}(\mathcal{D}, \mathbf{m}) = \underbrace{\sum_{\mathbf{c} \in \bigcup_{.=1}^{f} \mathcal{D}_1 \cup \bigcup_{.=1}^{f'} \mathcal{D}_2 \cup \bigcup_{.=1}^{f''} \mathcal{D}_3} |\texttt{tok}_\uparrow[\mathbf{m}](\mathbf{c})|}_{=(6f+6f'+4f'')J} + \underbrace{\sum_{\mathbf{c} \in \bigcup_{.=1}^{f'''} \mathcal{D}_4} |\texttt{tok}_\uparrow[\mathbf{m}](\mathbf{c})|}_{\geq (2J-1)2f'''+3f'''} + \underbrace{\sum_{\mathbf{c} \in \mathcal{D}_5} |\texttt{tok}_\uparrow[\mathbf{m}](\mathbf{c})|}_{>0} \tag{66a}$$

$$> (6f + 6f' + 4f'')J + (4J + 1)f''' \qquad \text{By construction } f''' = 5I \tag{66b}$$

$$= (6f + 6f' + 4f'' + 4f''') J + 5 I \tag{66c}$$

which contradicts the proof's statement. $\qquad\qquad\square$

**LemmaProofStep 5.** (Step $\text{⑤}$). *Any instance of the tokenisation problem with a solution which compresses the text by at least $(6f + 6f' + 6f'' + 6f'')J + 2I + \gamma$ symbols must be produced by a* max-2-SAT *problem with at least $\gamma$ satisfied clauses, i.e.,:*

$$\left( \texttt{toklen}(\mathcal{D}, \mathcal{S}) \leq \underbrace{(6f + 6f' + 4f'' + 4f''')J + 3I - \gamma}_{\sum_{\mathbf{c} \in \mathcal{D}} |\mathbf{c}| - (6f+6f'+6f''+6f''')J-2I-\gamma} \right) \implies \mathrm{M2S}(\mathcal{X}, \mathcal{C}, \gamma) \tag{67}$$

*Proof.* Finally, we now know any solution with this compression must have—for any variable $X_j$—either two merges in $\mathbf{m}_j^{\mathrm{T}}$ or in $\mathbf{m}_j^{\mathrm{F}}$ (and never both). We can thus create a bijection $\mathrm{Conv}_{\mathbf{m} \to \chi}$ between the set of possible merge sequences respecting this condition, and the set of T/F assignments to SAT variables $\chi$:

$$\mathrm{Conv}_{\mathbf{m} \to \chi}(\mathbf{m}) = \left\{ \begin{array}{ll} \mathtt{T} & \text{if } |\mathbf{m} \cap \mathbf{m}_j^{\mathrm{T}}| = 2 \\ \mathtt{F} & \text{if } |\mathbf{m} \cap \mathbf{m}_j^{\mathrm{F}}| = 2 \end{array} \right\}_{j=1}^{J} \tag{67}$$

Further, note that merge sequences of this form (as shown in Eq. (41)) lead to exactly $(6f + 6f' + 6f'' + 6f''')J$ symbols being compressed in datasets $\mathcal{D}_1$ to $\mathcal{D}_4$. To achieve the target compression, a solution must thus compress $\mathcal{D}_5$ by at least $2I + \gamma$ symbols. Now note that for any string in $\mathcal{D}_5$, e.g., $\odot x_j^{\mathrm{T}} \odot x_{j'}^{\mathrm{F}} \odot$, we have three compression options: $\odot x_j^{\mathrm{T}} \odot$ and $x_{j'}^{\mathrm{F}} \odot$ will be compressed, saving 3 symbols; $\odot x_j^{\mathrm{T}}$ and $\odot x_{j'}^{\mathrm{F}} \odot$ will be compressed, also saving 3 symbols; or only $\odot x_j^{\mathrm{T}}$ and $x_{j'}^{\mathrm{F}} \odot$ will be compressed saving only 2 symbols. More specifically, $\odot x_j^{\mathrm{T}} \odot$ will be compressed to a single symbol if merge $\langle \odot, x_j^{\mathrm{T}} \odot \rangle$ exists; similarly, $\odot x_{j'}^{\mathrm{F}} \odot$ will be compressed to a single symbol if merge $\langle \odot x_{j'}^{\mathrm{F}}, \odot \rangle$ exists. They cannot both be compressed, however, as there is only one symbol $\odot$ between the literals. We thus get a reduction of 3 symbols for each of these strings if at least one of its literals has an associated merge in $\mathbf{m}$. Note thus that whenever a string $\odot x_j^{\mathrm{T}} \odot x_{j'}^{\mathrm{F}} \odot$ is compressed by 3 symbols using merges $\mathbf{m}$, the max-2-SAT disjunction $X_j \vee \neg X_{j'}$ will also be satisfied by assignment $\chi = \mathrm{Conv}_{\mathbf{m} \to \chi}(\mathbf{m})$; similarly, whenever this string is only compressed by two symbols, the max-2-SAT disjunction will not be satisfied. A similar logic applies to all potential strings in $\mathcal{D}_5$: $\odot x_j^{\mathrm{T}} \odot x_{j'}^{\mathrm{F}} \odot$, $\odot x_{j'}^{\mathrm{T}} \odot x_j^{\mathrm{F}} \odot$, $\otimes x_j^{\mathrm{F}} \odot x_{j'}^{\mathrm{F}} \odot$, and $\odot x_j^{\mathrm{T}} \odot x_{j'}^{\mathrm{T}} \otimes$. As our condition assumes a compression of at least $2I + \gamma$ symbols, we know that we have at least $\gamma$ strings for which a literal has an associated merge. Now, let $\mathbf{m}^\star \in \mathcal{M}^\star$ be a valid solution to $\mathrm{Tok}_\uparrow(\mathrm{R2}(\mathcal{X}, \mathcal{C}, \gamma))$ and

$\chi^{\star} = \mathrm{Conv}_{\mathbf{m}\to\chi}(\mathbf{m}^{\star})$ the equivalent $\mathtt{max{-}2{-}SAT}$ assignment. We can thus write:

$$2I + \gamma \leq \sum_{\mathbf{c}\in\mathcal{D}_5} |\mathbf{c}| - |\mathtt{tok}_{\uparrow}[\mathbf{m}^{\star}](\mathbf{c})| \tag{68a}$$

$$=2I + \sum_{\mathbf{c}\in\mathcal{D}_5} \mathbb{1} \left\{ \begin{array}{c} \left(x_j^{\mathtt{T}} \in \mathbf{c}\right) \text{ and } \left(|\mathbf{m}_j^{\mathtt{T}} \cap \mathbf{m}^{\star}| = 2\right) \\ \text{or} \\ \left(x_j^{\mathtt{F}} \in \mathbf{c}\right) \text{ and } \left(|\mathbf{m}_j^{\mathtt{F}} \cap \mathbf{m}^{\star}| = 2\right) \\ \text{or} \\ \left(x_{j'}^{\mathtt{T}} \in \mathbf{c}\right) \text{ and } \left(|\mathbf{m}_{j'}^{\mathtt{T}} \cap \mathbf{m}^{\star}| = 2\right) \\ \text{or} \\ \left(x_{j'}^{\mathtt{F}} \in \mathbf{c}\right) \text{ and } \left(|\mathbf{m}_{j'}^{\mathtt{F}} \cap \mathbf{m}^{\star}| = 2\right) \end{array} \right\} \tag{68b}$$

$$=2I + \sum_{i=1}^{I} \mathbb{1}_{\chi^{\star}}\{L_i^1 \vee L_i^2\} \tag{68c}$$

$$\implies \mathrm{M2S}(\mathcal{X},\mathcal{C},\gamma) \tag{68d}$$

where $\mathbb{1}_{\chi^{\star}}\{L_i^1 \vee L_i^2\}$ evaluates $L_i^1 \vee L_i^2$ using assignments $\chi^{\star}$. We thus know that, if a satisfying tokenisation solution exists, then the associated $\mathtt{max{-}2{-}SAT}$ problem will also be satisfiable. This concludes the proof. $\qquad\square$

