# OpenReview forum: "Tokenisation is NP-Complete"
_ICML.cc/2025/Workshop/TokShop — TokShop_

### Official Review · Reviewer_VAV5 · 2025-06-03
**Tokenisation is NP-Complete.**

**Rating:** 8
**Confidence:** 4

**Review:**

**Summary**
This paper investigates the computational complexity of tokenization, defining it as the problem of compressing a dataset to a specified number of symbols. The authors define two primary variants: direct tokenization, which involves directly choosing a vocabulary of subwords, and bottom-up tokenization, which relies on a sequence of merge operations. The core contribution is proving that both of these variants are NP-complete, specifically by reducing them from the maximum 2-satisfiability (max-2-SAT) problem. The paper details the definitions of these tokenization problems, proves their inclusion in the NP class, and then demonstrates their NP-hardness through reductions. The practical implication of these findings is that optimal tokenization is likely intractable, reinforcing the current reliance on approximate algorithms in natural language processing. The paper also discusses variations such as deduped datasets and single long strings, and briefly touches upon the relationship between tokenization and dictionary compression.

**Strengths**
* Clear Definitions: The paper provides clear and formal definitions for direct and bottom-up tokenization, including their objective functions and decision problems. This clarity is crucial for understanding the subsequent complexity proofs.
* Rigorous Proofs (Sketches): The paper offers detailed proof sketches for both NP-completeness results, relying on reductions from the known NP-hard max-2-SAT problem. The appendices, though not fully provided in this extract, are referenced for formal proofs, indicating a thorough approach.
* Practical Implications: The authors effectively translate their theoretical findings into practical implications for the NLP community, suggesting a continued focus on approximate algorithms for tokenizer design.

**Weaknesses**
* Reliance on Appendix for Formal Proofs: While proof sketches are provided, the full formal proofs are relegated to the appendices, which are not included in the provided text. This makes it difficult to fully evaluate the rigor and correctness of the proofs without access to the complete document.
* Clarity of Reduction Construction: While the steps for constructing the dataset D and choosing K and δ in Reduction 1 and Reduction 2 are provided, understanding the exact mapping and the intricate details of how satisfying a tokenization instance corresponds to satisfying a max-2-SAT instance requires careful tracing of the variables f,f', etc and their roles, which can be challenging to grasp quickly.

---

### Official Review · Reviewer_ebBg · 2025-06-07
**The paper states the tokenization problem is NP-Complete, based on the equivalence between itself and the Maximum 2-Satisfiability problem.**

**Rating:** 10
**Confidence:** 4

**Review:**

Summary:
The paper formally presents direct tokenization (based on splitting into subwords) and bottom-up tokenization (based on merging characters into subwords). Creating a tokenizer that compresses a dataset to a maximum of N symbols is equivalent to deciding how many clauses are satisfiable in the Maximum 2-satisfiability problem. The latter problem, being NP-Complete, proves that finding the best tokenizer is on the same spectrum.

Strengths:
- The presentation format is rigorous and consistent. The color notation helps the reader follow the formal content more easily.
- The authors provide consistent proof of the equivalence between the tokenization problems and the Maximum 2-Satisfiability problem.
The problem is relevant to improving tokenizers for the most recent LLMs, most of which are based on the bottom-up paradigm.

Weakness:
- Several footnotes present related work, but maybe it is better to have a separate section.

Typos and revisions:
- In the color convention, several notations are pink symbols followed by black indices, e.g., $s_1$ and $s_2$ on line 134. Shouldn't it look more consistent with both pink colors?
- Formulas 12a and 20 lack the variable that starts from 1 and goes to $f$ or $f'$.

---

### Official Review · Reviewer_R49S · 2025-06-09
**Theoretically sound and interesting, but with little practical impact**

**Rating:** 6
**Confidence:** 3

**Review:**

The paper shows NP-completeness of two types of compression-optimal tokenizers: direct and bottom-up.
(BPE is the most popular example of greedy-approximation bottom-up tokenizer optimizing for compression.
SentencePiece UnigramLM is an example of a greedy direct tokenizer, but optimizing unigram log-probability, not compression.)

=== Strengths ===

The paper introduces a nice theoretical framework/terminology for studying tokenizers.
I really like the clarity and exactness of presentation (including the colorful notations).
I could not find any weaknesses in the proofs.

=== Weaknesses ===

The paper assumes all tokenizers are motivated by an objective function G,
but only compression utility is studied in the paper.
There are many implementations where it would be difficult to find any G,
e.g. T2T's SubwordTextEncoder (WordPiece) pretokenizes on alphanumeric/non-alphanumeric boundaries
and omits single-space tokens, which results in a non-concatenative tokenizer,
so Formulas (1) and/or (2) do not apply.

Another possible G---Unigram log-probability---is mentioned,
but not studied (as admitted in the Limitations section).

Footnote 2 mentions two "related concurrent works", both pre-prints,
but the one from 2024 already proves "something stronger: its APX-hardness" for the bottom-up tokenizers.
The other proves "NP-completeness of a restricted variant of direct tokenisation,
in which a set of candidate tokens is previously specified".
So even from the theoretical point of view, it seems that the novel part of this paper
is only the proof of direct tokenizers
(but with another restriction - no fixed alphabet size).

It is not clear what is the practical impact of the paper for the following 4 reasons.

1) No example of a direct tokenizer optimizing compression utility is given in the paper.

2) As mentioned in the Limitations section, most (multilingual) tokenizers use bytes as the input symbols (instead of Unicode characters), i.e. a fixed alphabet, for which the theoretical results do not apply, so "more efficient algorithms might exist".

3) While Kozma & Voderholzer (2024) show a worst-case artificial example where the compression utility of BPE is only 0.625,
it is not what is the empirical compression utility range on real-world texts.

4) Footnote 1 says that "The compression achieved by a tokeniser
correlates with downstream language modelling performance".
Footnote 4 admits that "Schmidt et al. (2024) [...] argue
that compression and downstream performance have a more
complex relationship than prior work suggests".
Schmidt et al. (2024) actually write
"Through extensive experimentation we find this hypothesis
[that fewer tokens lead to better downstream performance]
not to be the case".
While this paper cites Schmidt et al. (2024) and their PathPiece method,
it does not reflect their results (e.g. in the Limitations section).


=== Minor comments ===

line 072: K has not been defined yet.
I think the whole "we say its size is |S| = |Σ| + K" can be omitted here
(whereas this introductory part of Section 3 should be
relevant for both bottom-up and direct tokenization);
it is enough that it is mentioned in Formula (8).

line 434: "Relatedly, in practice, state-of-the-art language models still
achieve very good performance (at least in English), despite sub-
optimal tokenisers."
Suboptimal in which sense?
Intrinsically, i.e. regarding the compression,
"greedy BPE may be close to optimal" (Zouhar et al., 2023b).
Extrinsically, i.e. regarding the performance in downstream tasks,
there are many hints that better tokenizers may help, especially
in multilingual settings, but no such tokenizers are used in practice
instead of BPE/UnigramLM yet.
That said, I agree with the main message of this footnote 13.

---

### Decision · Program_Chairs · 2025-06-10

Accept